# Kinetic analysis of ASIC1a delineates conformational signaling from proton-sensing domains to the channel gate

**Sabrina Vullo[1], Nicolas Ambrosio[1], Jan P Kucera[2], Olivier Bignucolo[1,3], Stephan Kellenberger[1]***

[1]Department of Biomedical Sciences, University of Lausanne, Lausanne, Switzerland; [2]Department of Physiology, University of Bern, Bern, Switzerland; [3]SIB, Swiss Institute of Bioinformatics, Lausanne, Switzerland

**Abstract** Acid-sensing ion channels (ASICs) are neuronal Na$^+$ channels that are activated by a drop in pH. Their established physiological and pathological roles, involving fear behaviors, learning, pain sensation, and neurodegeneration after stroke, make them promising targets for future drugs. Currently, the ASIC activation mechanism is not understood. Here, we used voltage-clamp fluorometry (VCF) combined with fluorophore-quencher pairing to determine the kinetics and direction of movements. We show that conformational changes with the speed of channel activation occur close to the gate and in more distant extracellular sites, where they may be driven by local protonation events. Further, we provide evidence for fast conformational changes in a pathway linking protonation sites to the channel pore, in which an extracellular interdomain loop interacts via aromatic residue interactions with the upper end of a transmembrane helix and would thereby open the gate.

## Introduction

This study investigates the activation mechanism of acid-sensing ion channels (ASICs), a family of H$^+$-gated Na$^+$ channels of the nervous system (*Waldmann et al., 1997*; *Wemmie et al., 2013*; *Kellenberger and Schild, 2015*; *Yang and Palmer, 2014*). ASIC activation is linked to physiological and pathological processes such as learning and pain sensation, neurodegeneration after ischemic stroke, fear, and anxiety (rev. in *Wemmie et al., 2013*; *Kellenberger and Schild, 2015*). ASICs respond to extracellular acidification with a transient current, because after opening, they enter a non-conducting desensitized state (*Waldmann et al., 1997*; *Gründer and Pusch, 2015*). High-resolution structures of chicken ASIC1 (cASIC1a) in the closed (*Yoder and Gouaux, 2020*; *Yoder et al., 2018*) (and (*Sun et al., 2020*), human ASIC1a), toxin-opened (*Baconguis et al., 2014*; *Baconguis and Gouaux, 2012*; *Dawson et al., 2012*), and desensitized conformation (*Gonzales et al., 2009*; *Jasti et al., 2007*) are available. Functional ASICs are trimers (*Bartoi et al., 2014*). Each ASIC subunit consists of short intracellular *N*- and *C*-terminal ends, two transmembrane domains TM1 and TM2, and a large extracellular region, with the shape of a hand, organized in defined domains that have been named palm, knuckle, β-ball, thumb, and finger (*Figure 1A*).

Each ASIC channel contains three 'acidic pockets' – regions with a high density of acidic residues – which are enclosed by the thumb, finger and β-ball of one, and the palm of a neighboring subunit. The lower palm domains enclose the central vestibule. The wrist links the extracellular channel parts to the transmembrane segments. The acidic pocket, the palm and the wrist are pH-sensing regions that are potentially involved in ASIC activation (*Paukert et al., 2008*; *Krauson et al., 2013*; *Liechti et al., 2010*; *Vullo et al., 2017*; *Schuhmacher et al., 2015*). It is expected that protonation would locally induce structural rearrangements, and these conformational changes would be

*For correspondence:
Stephan.Kellenberger@unil.ch

**Competing interests:** The authors declare that no competing interests exist.

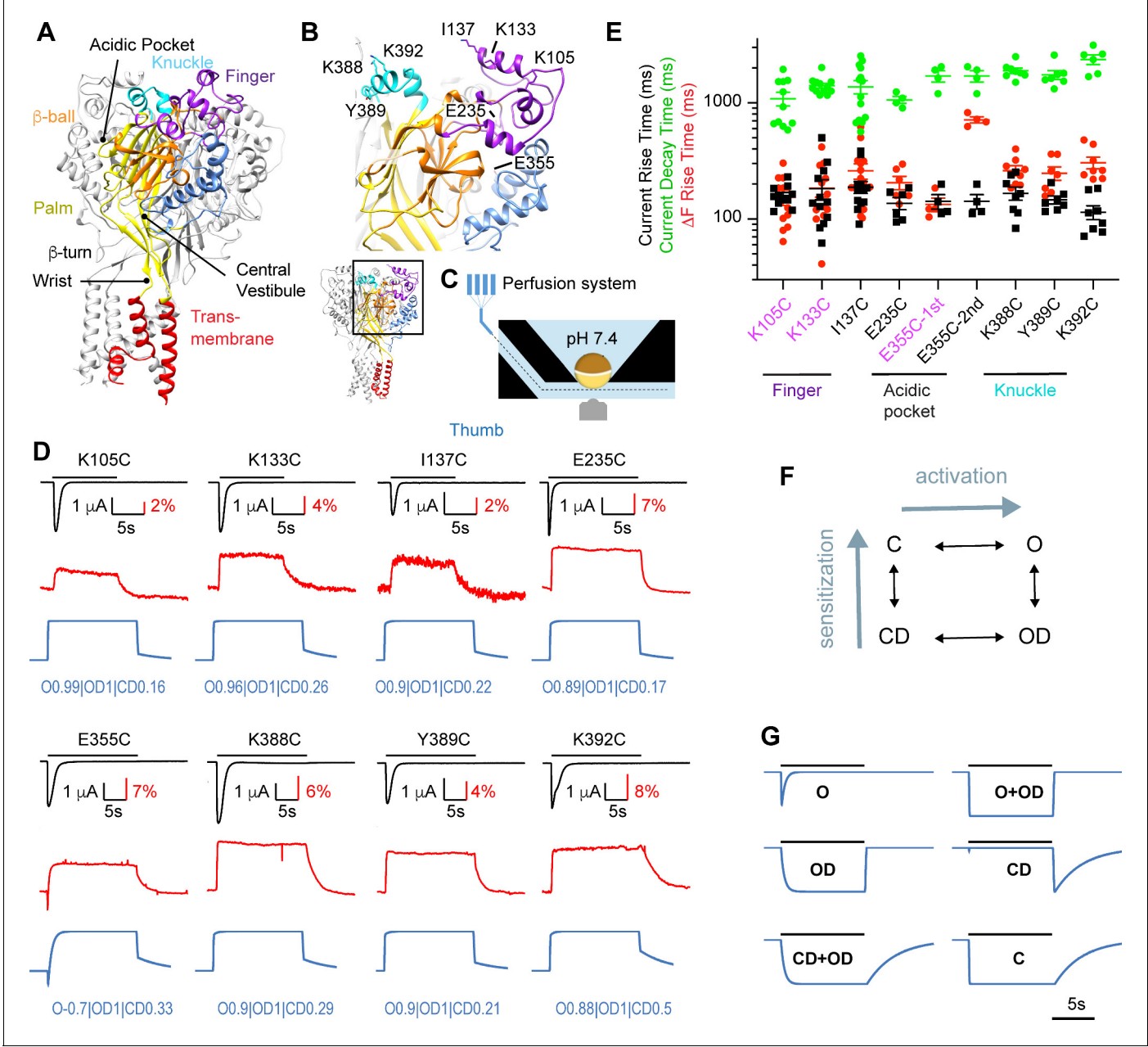

**Figure 1.** Rapid ΔF signals of mutants distant from the pore. (A) Structural image of a trimeric ASIC1a channel with the domains colored and named, based on the Mit Toxin-opened structure (PDB identifier 4NTW, *Baconguis et al., 2014*); the acidic pocket, central vestibule, β-turn, and wrist are indicated. (B) Detailed structural view with indication of residues studied in Figure 1. (C) Schematic view of the oocyte chamber used for the kinetic measurements of currents and fluorescence signals (ΔF). (D) Current traces (black), fluorescence signals (red) and simulated fluorescence traces (blue) are shown. The conditioning pH between stimulations was pH7.4; the stimulation pH6 was applied as indicated by the horizontal black bars. Simulated traces were generated by the kinetic model (Materials and methods), using the parameters indicated in blue (*Supplementary file 2*). (E) Kinetics of current appearance and decay, and of ΔF onset, measured as rise time (RT) or decay time for an activation by pH6.0 (n = 4–14). An analysis was carried out to determine if the ΔF onset kinetics were correlated with current appearance or decay (Materials and methods). In mutants labeled in purple, the ΔF onset kinetics were correlated with current appearance. For E355C, ΔF onset kinetics are reported for both components ('1st' and '2nd'). (F) Representation of the kinetic model used for the simulation of fluorescence traces; C, closed; O, open; OD, open-desensitized; CD, closed-desensitized. (G) Model-generated fluorescence with proportionality factor = -1 for the state(s) indicated with the traces; for C, this factor was +1. Conditioning pH was 7.4; it was changed to pH6.0 for a duration of 10 s. The fall time of the pH change (speed of solution change) was set to 200 ms. Source data are provided in the file *Figure 1—source data 1*.

The online version of this article includes the following source data and figure supplement(s) for figure 1:

**Source data 1.** Mutants distant from the pore.

*Figure 1 continued on next page*

*Figure 1 continued*

**Figure supplement 1.** Kinetics of mutants with rapid ΔF located outside the wrist at pH6.5 and 5.5.

transmitted to the gate. It may also be possible that the opening of the ASIC1a gate is controlled by protonation events in the wrist alone, since combined mutation of two putative pH-sensing His residues in the wrist suppressed ASIC1a activation, while maintaining cell surface expression intact (*Paukert et al., 2008*).

By using voltage-clamp fluorometry (VCF), we show here that conformational changes with the kinetics of ASIC activation occur in the wrist and in several distal extracellular parts, consistent with pH sensing in different channel regions. To investigate possible pathways transmitting conformational changes from distant sites to the channel gate, we determined the kinetics and direction of conformational changes in the palm and the adjacent palm-thumb loops, using VCF combined with the introduction of fluorescence quencher groups. The fluorescence signals were analyzed with a kinetic model to further support the attribution of fluorescence changes to specific functional transitions. This analysis allowed us to propose sequences of conformational rearrangements in ASIC domains in the proximity of the β1-β2 linker during activation and desensitization. Inspection of the crystal structures, and unbiased Molecular Dynamics (MD) simulations were used to interpret the VCF predictions.

## Results

### Fast conformational changes in ASIC domains that are distant from the pore

The aim of a first set of experiments was to compare the kinetics of conformational changes in different parts of ASIC1a. To this end, VCF was used, which measures simultaneously with the channel current, changes in fluorescence (ΔF) of a strategically placed fluorophore, as a readout of changes in its environment. ASIC1a constructs containing engineered Cys residues for the docking of maleimide derivatives of fluorophores were expressed in *Xenopus laevis* oocytes, and oocytes were exposed to maleimide derivatives of AlexaFluor488 or CF488A prior to the measurement. The ΔF signals detected in labeled mutants are due to fluorophores attached to the engineered Cys residues, since exposure of wild-type (WT) ASIC1a to the maleimide derivatives of these two fluorophores does not lead to fluorescence changes upon acidification (*Bonifacio et al., 2014*).

In previous studies, we had observed rapid ΔF signals with mutations located distantly from the channel gate (*Bonifacio et al., 2014*; *Gwiazda et al., 2015*; *Vullo et al., 2017*). For an accurate comparison of the kinetics of these mutants (*Figure 1B*), they were measured here in a measuring chamber in which the current and ΔF signal are measured from approximately the same oocyte surface (*Figure 1C*, *Vullo et al., 2017*). The tested mutants produced transient currents upon extracellular acidification (black traces in *Figure 1D*). The ΔF signals of these mutants (red traces) were sustained; only E355C showed an additional first transient component. *Figure 1E* presents for acidification to pH6 the rise time (time to pass from 10% to 90% of the maximal amplitude, RT) of the ΔF onset (red symbols) and of the current appearance (black), and the decay time (90% to 10%) of the current desensitization (green). The kinetic analysis illustrates that for most of these mutants, the ΔF onset kinetics are similar to those of current appearance (*Figure 1E*). To determine how closely the conformational changes underlying the ΔF signals were associated with a functional transition, a correlation analysis was done for each mutant (Materials and methods). This indicated a correlation to channel activation for K105C, K133C, and E355C (purple label). The results obtained at pH6.5 and 5.5 (*Figure 1—figure supplement 1*) were comparable with those at pH6.0 shown here.

The ΔF signals indicate that the environment of the docked fluorophores changes, thus that the acidification induces conformational changes. The ΔF may be due to changes in solvent exposure of the fluorophore (*Gandhi and Olcese, 2008*; *Cha and Bezanilla, 1997*; *Mannuzzu et al., 1996*), or to changes of the exposure to nearby quenching groups (*Pantazis and Olcese, 2012*; *Vullo et al., 2017*). It is expected that conformational transitions that are not directly associated with a defined functional state also influence the fluorescence signal. As an additional strategy for testing the association of ΔF signals with specific transitions, ΔF traces were compared to traces generated by a

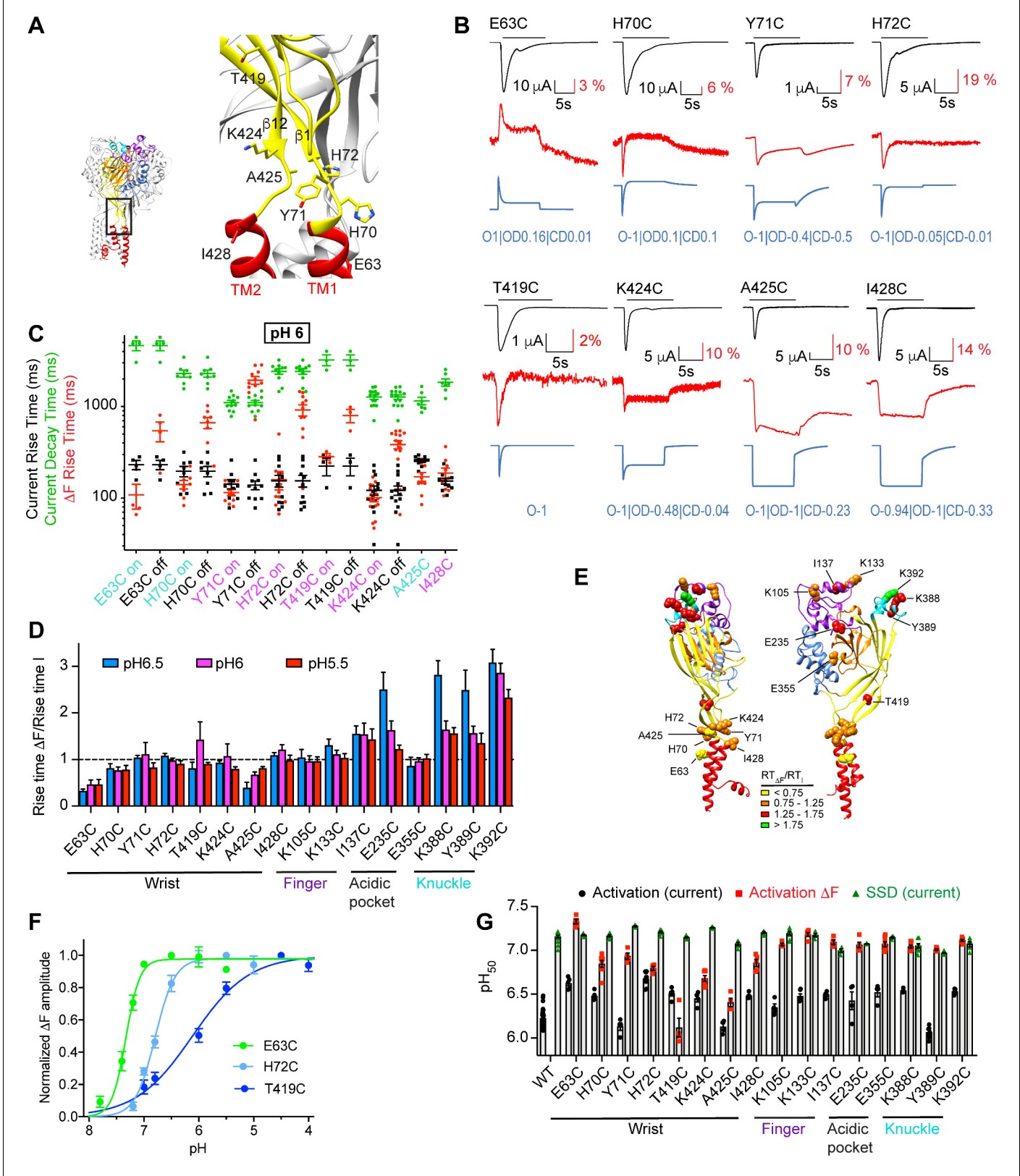

**Figure 2.** Transient, rapid ΔF signals in the wrist. (**A**) Structural image indicating the positions of the tested mutants. (**B**) Representative current and fluorescence traces of the Cys mutants in the wrist. The conditioning pH between stimulations was pH7.4; the stimulation pH6 was applied as indicated by the horizontal black bars. The blue traces were generated by the kinetic model as described in the legend to *Figure 1G*. (**C**) The kinetics of current

*Figure 2 continued on next page*

*Figure 2 continued*

appearance and decay, and of ΔF onset, measured as rise time (RT) or decay time for an activation by pH6.0, are plotted (n = 3–14). Since the ΔF signal was transient in many mutants, the kinetics of the ΔF onset ('on') and decay ('off') are indicated. The color of the labels of the mutants indicates that the ΔF kinetics are correlated with current appearance (purple) or are faster than current appearance (cyan). (**D**) The $RT_{\Delta F}/RT_I$ ratio is plotted for stimulation pH6.5, 6.0, and 5.5 (n = 3–14) of wrist and distant mutants. (**E**) Structural image of ASIC1a (seen from two different sides) indicating the position of the mutants studied in *Figures 1* and *2*. The color of each residue corresponds to a range of $RT_{\Delta F}/RT_I$ ratio at pH6, as indicated in the figure. (**F**) Experimentally determined pH dependence of the ΔF amplitude for the indicated mutants; n = 4–5. (**G**) Summary of pH dependencies of current activation (black), current SSD (green) and ΔF amplitude (red), n = 4–30. In the experiments for ΔF measurements, the conditioning pH was 8.0. Source data are provided in the file *Figure 2—source data 1*.

The online version of this article includes the following source data and figure supplement(s) for figure 2:

**Source data 1.** Wrist mutants.
**Figure supplement 1.** Rapid ΔF kinetics of wrist mutants at pH5.5 and 6.5.
**Figure supplement 2.** pH dependence of mutants involved in rapid conformational changes.

kinetic model under the assumption that each functional state contributes with a proportionality factor that lies between −1 and +1 to the measured fluorescence. To this end, a kinetic ASIC1a model recently developed in our laboratory was used (*Alijevic et al., 2020*), which is based on the Hodgkin-Huxley formalism containing an activation and a sensitization gate. The model contains four states, an open (O), a closed (C) and two desensitized states, closed-desensitized (CD) and open-desensitized (OD; *Figure 1F*). This model reproduces the acid-induced ASIC1a activity well (*Alijevic et al., 2020*). *Figure 1G* shows ΔF traces generated by this model for a pH change from 7.4 to 6.0, under the assumption that in each case one single functional state, or a pair, O+OD (corresponding to the activation gate) and CD+OD (sensitization gate) contributes to the signal. The ΔF onset is most rapid for the fluorescence associated with the O or O+OD states. The CD state is associated with a substantial ΔF only at the return to pH7.4 (*Figure 1G*). At the end of the acidic pulse when the pH is changed back to 7.4, the return to the initial fluorescence level is slow if the fluorescence is associated with the CD state, consistent with the slow recovery from desensitization at pH7.4 (*Alijevic et al., 2020*; *Rook et al., 2020*).

By attributing proportionality factors between −1 and +1 to the four states of the model, based on the shape and the kinetics of experimental ΔF traces (Materials and methods, experimental values of the ΔF kinetics at the return to the conditioning pH are provided in *Supplementary file 1*), ΔF signals can be generated that closely resemble the measured ΔF signals, as illustrated by the blue traces shown in *Figure 1D*, where the proportionality factors (*Supplementary file 2*) are indicated for each mutant. This model suggests high association of the ΔF signal with the O and OD state, and lower association with the CD state.

## Fast conformational changes in the wrist have the same timing as channel opening

It was then tested whether the ΔF kinetics were different if the fluorophores were placed closer to the channel gate. Mutations were therefore introduced in the wrist (*Figure 2A*). In contrast to the mutants at distant positions (*Figure 1*), the ΔF traces of these mutants had two components, a first transient and a second sustained part (*Figure 2B*). For transient ΔF components, the RT of the onset ('on') and of the decay of the signal ('off') were determined. The analysis shows that the kinetics of the fast ΔF component of all mutants were equal to or faster than current appearance (*Figure 2C–D*, Materials and methods; pH6.5 and 5.5 data in *Figure 2—figure supplement 1*). The traces were best modeled with highest proportionality factors for the open state, highlighting the association of the ΔF of these mutants with channel opening.

To compare the ΔF properties of these mutants with mutations located distantly from the gate, the RT of the ΔF onset was in each experiment normalized to the RT of the current appearance; a ratio <1 indicates that the ΔF kinetics are faster than the kinetics of current appearance. The $RT_{\Delta F}/RT_I$ ratio, determined at pH6.5, 6.0 and 5.5, was in the range of ~0.5–1.3 for all wrist mutants (*Figure 2D*). The $RT_{\Delta F}/RT_I$ ratios are visualized in a color code in the structural image in *Figure 2E*. The three tested mutants of the finger (K105C, K133C, and I137C) and E355C of the acidic pocket displayed also $RT_{\Delta F}/RT_I$ ratios close to 1, while this ratio was higher in the acidic pocket mutant

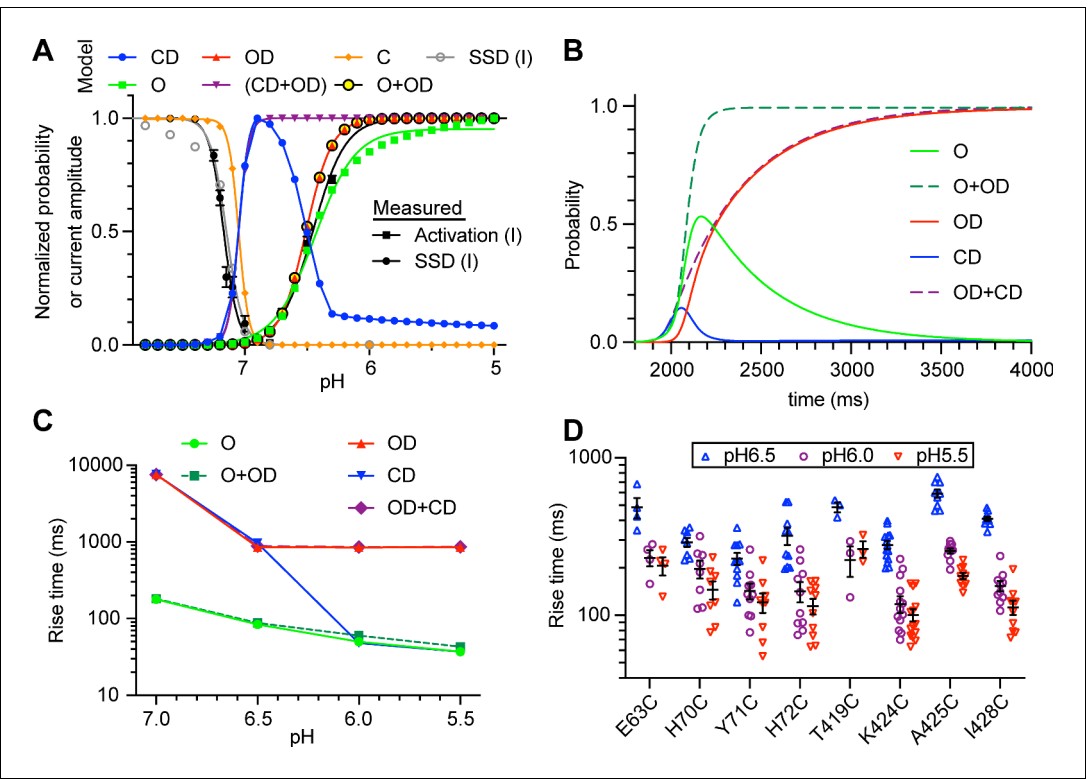

**Figure 3.** Analysis of pH dependence and kinetics of ΔF signals indicate that rapid ΔF signals are linked to channel opening. (**A**) Model-generated normalized probability of occupancy of the indicated states (colored traces). The conditions of the simulations were as in *Figure 1G*, except that the duration of the acidic pulse was 20 s. The gray symbols represent the current SSD calculated with the model by applying the same protocol as applied in the experiments (*Figure 2—figure supplement 2*). The black symbols show the experimental pH dependence of current activation and SSD as determined in *Figure 2—figure supplement 2*. (**B**) Model-generated time dependence of state occupation upon a pH change from 7.4 to 6.0. Conditioning pH was 7.4; it was changed to pH6.0 at 2 s for a duration of 10 s. The fall time of the pH change was set to 200 ms. (**C**) Model-generated rise time of the occupancy of the indicated states upon acidification from pH7.4 to the indicated values; model parameters as in B. (**D**) Experimentally determined rise time of the ΔF signal measured for the wrist mutants at the three indicated pH values. Source data are provided in the file *Figure 3—source data 1*.

The online version of this article includes the following source data and figure supplement(s) for figure 3:

**Source data 1.** Fluorescence kinetics of wrist mutants at pH 6.5, 6.0 and 5.5.

**Figure supplement 1.** Modeled state probability at different pH conditions.

---

E235C and in the knuckle mutants (K388C, Y389C, K392C) compared to most mutants localized in the wrist at pH6.5 (Two-way ANOVA, Sidak post-test, p<0.05).

The pH dependence of current activation and steady-state desensitization (SSD, corresponding to the closed-desensitized transition), and of the ΔF amplitude were determined (*Figure 2F*, *Figure 2—figure supplement 2*), and the pH values of half-maximal effect, pH50, are presented in *Figure 2G*. This shows that even for mutants whose ΔF was associated with channel opening, based on the kinetics and the kinetic model, the pH50 of the ΔF signal was in most cases more alkaline than the pH50 of current activation, and was for several mutants close to the pH50 of SSD. Possible reasons for this shift are discussed below (*Discussion*).

## Fast ΔF signals are associated with channel opening

When the pH is changed from 7.4 to an acidic value, the channel can move from C to either O or CD (*Figure 1F*). To estimate whether the measured fast ΔF signals might be associated with the C-CD transition (i.e. whether the CD state contributes to this fluorescence), the above-mentioned kinetic model of ASIC1a function (*Alijevic et al., 2020*) was used to predict the pH dependence of the

probability of these different states and of the kinetics of their appearance. *Figure 3A* plots, based on a step acidification, the normalized peak probability of the four states and of the two gates (O +OD and CD+OD) for the indicated pH range. As expected, the normalized peak probability of the CD state is biphasic (blue in *Figure 3A*), because of a strong occupancy of the C state at pH $\geq 7$, and of the OD state at pH <6.5. Calculated probability time courses of the different states at different pH conditions further illustrate this biphasic pH dependence of the CD state (*Figure 3—figure supplement 1*).

The pH dependence of the normalized peak probability of CD, CD+OD, and C is shifted alkaline with regard to that of the other states, and is close to the pH dependence of SSD, shown as black and gray symbols for experimental and model data, respectively. The time dependence of the probability of these states for an acidification from pH7.4 to 6.0 (*Figure 3B*) indicates a rapid increase in the probability of O, O+OD, and CD upon acidification. Slow onset kinetics are observed with OD and OD+CD. These kinetics show a certain acceleration with the acidity of the stimulation (*Figure 3C*) that is also observed with experimental ΔF kinetics (*Figure 3D*). The model strongly suggests that the fast ΔF signals are associated with opening, since it indicates that a ΔF associated with the CD state would have a biphasic pH dependence, which is clearly not observed in the experiments (*Figures 2F–G* and *3A*). If associated with both the CD and the OD state, the kinetics would be considerably slower than the kinetics of channel opening (*Figure 3B*).

The presented data show so far that rapid conformational changes occur in the wrist and at sites distant from the pore upon acidification. The kinetic ASIC1a model supports the interpretation that these rapid signals are associated with channel opening. To obtain more information on how the rapid conformational changes occurring at the distant sites are transmitted to the gate, we have in the following investigated conformational changes in the palm and nearby domains.

## Slow approaching between the palm β1-β2 linker and the β12 strand

The ΔF signals of A81C, S83C, and Q84C, located in the β1-β2 linker, on top of the lower palm β strand 1 (*Figure 4A–B*), may be due to quenching by Tyr417, which is located on β strand 12. When Tyr417 was mutated to Val (which does not quench the fluorescence) in the background of these Cys mutants, the ΔF signal was strongly decreased in the background of A81C, S83C, and Q84C (*Figure 4B–C*), indicating that ~70% of the ΔF signal in the single mutants A81C and S83C, and ~54% in Q84C, was due to a change in distance or orientation relative to Tyr417. The negative ΔF of the single mutants indicates an increased quenching, thus an approaching between the fluorophore attached to the Cys mutants and Tyr417. The ΔF onset kinetics were intermediate between those of current appearance and current decay and in some conditions correlated with current decay (*Figure 4D*), indicating that the underlying conformational changes may be involved in the induction of ASIC desensitization. The $pH_{50}$ of the ΔF onset was close to that of SSD (*Figure 4—figure supplement 1*).

## Conformational changes in the β1-β2 linker and the β5-β6 loop precede desensitization

These double mutants were then used as a basis for placing Trp, a strong fluorescence quencher, at different positions in the proximity of the fluorophore docking positions A81C, S83C, or Q84C, to monitor additional distance changes. Studies with different fluorophores indicated quenching by Trp at distances $\leq 15$ Å (*Mansoor et al., 2002*; *Pantazis et al., 2018*). The β5-β6 loop of the β-ball runs almost parallel to the membrane surface, above the β1-β2 linker. In the direction from Pro205 to Met210, it approaches a subunit interface (*Figure 5A*; *Lynagh et al., 2018*; *Bignucolo et al., 2020*). Nine mutants were identified in which the placement of a Trp in the β5-β6 palm loop generated fluorescence signals (*Figure 5A–B*). Most triple mutants showed normal transient currents. Only those containing mutations of K208, T209 and M210, and L207 in the background of A81C/Y417V, had a slowed or disrupted desensitization (*Figure 5B–C*). The ΔF signals at pH6.0 were in most cases sustained and of a single component, except for the ΔF signals of Q84C/Y417V/P205W and A81C/Y417V/L207W, which appear to be the sum of an early negative and slightly delayed positive ΔF component (*Figure 5B*). A control stimulation protocol in which the channels are exposed to pH6.7 (which desensitizes the channels) before stimulation by pH6 was used to test whether a signal may potentially be non-specific (Materials and methods, *Supplementary file 3*). ΔF components

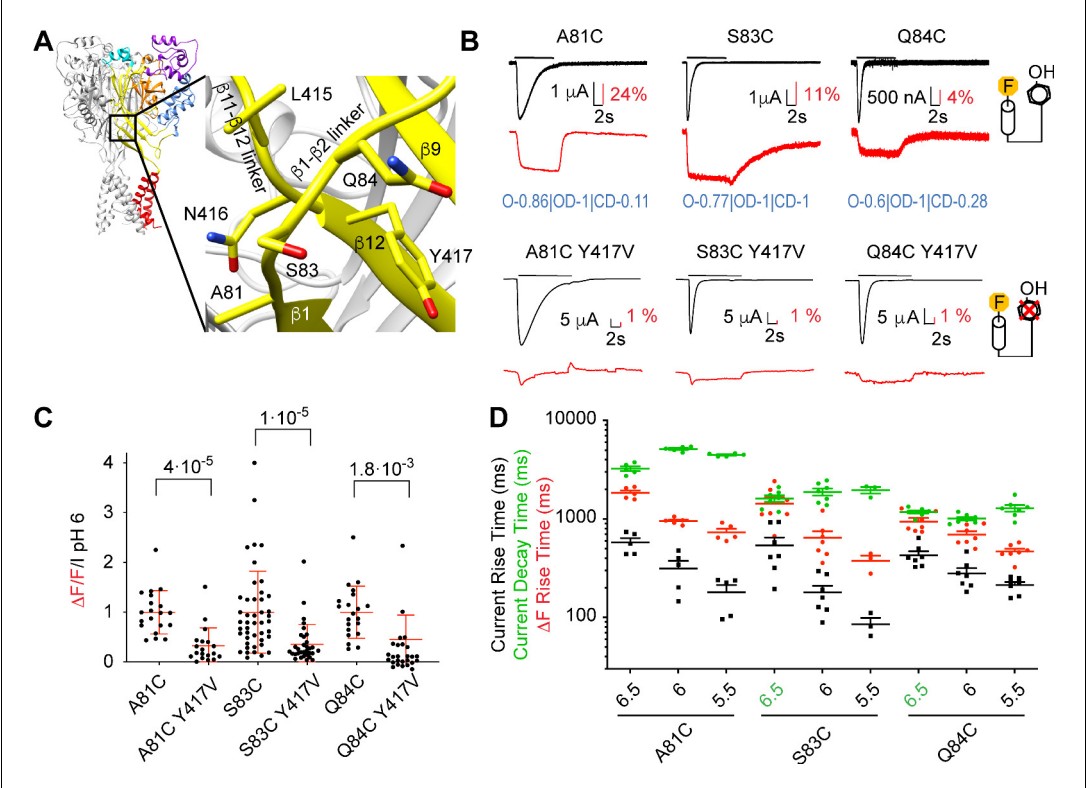

**Figure 4.** Removal of a fluorescence quenching group strongly reduces β1-β2 linker ΔF amplitude. (**A**) Structural image of ASIC1a (left) with zoom on the lower palm domain (right). (**B**) Representative current and fluorescence traces of single (top) and double (bottom) palm mutants. Conditioning pH7.4 was used, and mutants were stimulated by pH6 for the duration indicated by the horizontal bars. On the right, cartoons of a fluorophore and a quenching Tyr residue illustrate that in the double mutants the quencher was removed. Proportionality factors of the corresponding simulated traces are indicated in blue. (**C**) Normalized ratio of the fluorescence change / total fluorescence, divided by the pH6-induced current amplitude (ΔF/F/$I_{pH6}$) for single and double mutants; Kruskal-Wallis and Dunn's post-test, n = 19–45. Corresponding single and double mutants were measured on the same days, and for each experimental day, the ΔF/F/$I_{pH6}$ ratio of each cell of a given single/double mutant pair was divided by the average ΔF/F/$I_{pH6}$ ratio of the single mutant of that day. (**D**) Current RT and decay time and ΔF RT of single mutants obtained at pH6.5, 6.0 or 5.5 (n = 3–7). A green pH indication indicates that the ΔF onset kinetics are correlated with current decay. Source data are provided in the file **Figure 4—source data 1**.

The online version of this article includes the following source data and figure supplement(s) for figure 4:

**Source data 1.** Palm mutants.
**Figure supplement 1.** pH dependence of palm mutants.

identified as potentially non-specific are marked 'ns' in **Figure 5B** and were not further considered for the analysis. The ΔF kinetics of most mutants were slower than those of current appearance (**Figure 5C** and **Figure 5—figure supplement 1**), indicating that the underlying conformational changes are likely preparing or associated with desensitization. The ΔF traces of most mutants were well reproduced by the kinetic model, except for Q84C/Y417V/P205W and A81C/Y417V/L207W (**Figure 5—figure supplement 2**). Most mutants had highest proportionality factors with the OD, and smaller with the O state. The VCF analysis of these fluorophore-quencher pairs indicated thus nine distance changes, two of them associated with opening, seven preparing desensitization (**Table 1**). These findings were then used to obtain information on conformational changes, as shown further below.

## Detection of fast conformational changes in palm-thumb loops

To follow the kinetics of distance changes to close residues of a neighboring subunit, fluorophore docking at A81C, S83C, or Q84C, in the background of the Y417V mutation, was combined with positioning of the quenching group in three different sub-domains, at different distances from the plasma membrane (**Figure 6A**): (1) close to the β-turn ('T289W; the prefix ' indicates a residue of a

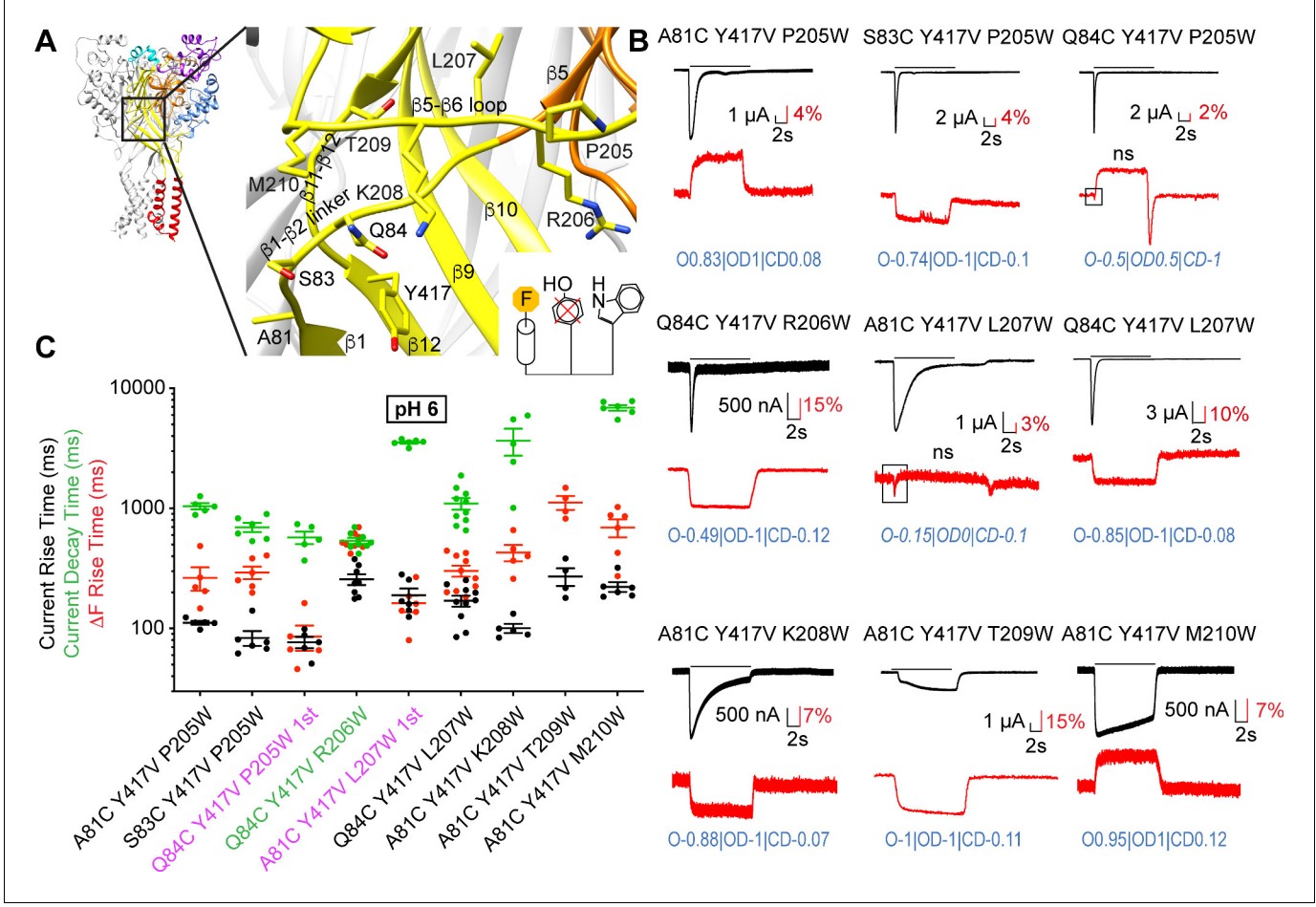

**Figure 5.** Slow structural rearrangements in the palm domain. (A) Structural image of ASIC1a (left) with a zoom on the palm (right), showing the position of mutations studied in this figure. Bottom, cartoon illustrating the approach used for these triple mutants, with the fluorophore ('F', on the left), the original quenching group Tyr417 (center) mutated, and a new quenching Trp (right) introduced at a different position. (B) Representative current and fluorescence traces of the triple mutants. Conditioning pH7.4 was used, and mutants were stimulated by pH6 for the duration indicated by the horizontal bars. The black frames in some ΔF traces highlight the first ΔF component. 'ns' indicates that this ΔF component was potentially non-specific (*Supplementary file 3*). Proportionality factors of the corresponding simulated traces are indicated in blue. (C) Current RT and decay time and ΔF RT obtained at pH6, n = 4–10. The color of the labels of the mutants indicates that the ΔF onset kinetics are correlated with current appearance (purple) or decay (green; Materials and methods). Source data are provided in the file *Figure 5—source data 1*.

The online version of this article includes the following source data and figure supplement(s) for figure 5:

**Source data 1.** Intrasubunit triple mutations palm.

**Figure supplement 1.** Slow ΔF kinetics of intrasubunit palm triple mutants at pH6.5 and 5.5.

**Figure supplement 2.** Model-generated ΔF patterns for two mutations of *Figure 5*.

neighboring subunit), (2) in the palm β10 strand ('L369W), and (3) at the end of the thumb helix α5 and in the linker between the thumb α5 helix and the palm β strand 10 ('D357W, 'Q358W, 'E359W; 'α5-β10 loop'). The distances compatible with quenching in these triple mutants are those between subunits (*Figure 6A*). These triple mutants produced normal transient pH-activated ASIC currents, with the exception of S83C/Y417V/D357W, which showed only a very small current with a transient and a sustained component (*Figure 6B*). The ΔF kinetics of four out of eight of these mutants were correlated with the kinetics of current appearance or were faster, indicating that at least one of the partners (S83/'T289, S83/'Q358, S83/'E359, A81/'L369) moves with the speed of activation (*Figure 6B–C* and *Figure 6—figure supplement 1*). Most ΔF traces could be well fitted with the kinetic model (*Figure 6—figure supplement 2*) indicating generally high proportionality factors for OD and O. This set of VCF measurements provided information on four distance changes occurring in the C-O transition, and six in the O-D transition.

**Table 1.** Summary of VCF analyses.

The polarity of the first or second ΔF signal components is indicated as negative (neg) or positive (pos), and the association of a ΔF signal component with either opening (o) or desensitization (d); 'i' stands for intermediate, that is between opening and desensitization. 'neg' and 'pos' are in regular font if no correlation with distance changes indicated by the structures is possible because the distance changes are too small (<1 Å), **bold** if they correlate with the structural prediction ('o' associated with closed-open distance change, 'i' and 'd' associated with open-desensitized distance change), and in *italics* if they contradict the structural predictions. *, these residue pairs were not used for structural interpretation; the pair A81-P205 because of the high distance (>20 Å, **Table 2**), and A81-T209 because the current of this mutant did not desensitize.

| Residue1 | Residue2 | VCF signal | | |
|----------|----------|------------|--------|-------------|
| | | Comp. | Signal | Association |
| Closed-to-open transition | | | | |
| Q84 | P205 | 1st | neg | o |
| A81 | L207 | 1st | neg | o |
| S83 | T289 | | **neg** | o |
| S83 | Q358 | 1st | **pos** | o |
| S83 | E359 | | *neg* | o |
| A81 | L369 | 1st | **neg** | < o |
| Open-to-desensitized transition | | | | |
| A81 | Y417 | | **neg** | i |
| S83 | Y417 | | **neg** | i |
| Q84 | Y417 | | neg | i |
| *A81 | P205 | | pos | i |
| S83 | P205 | | neg | i |
| Q84 | R206 | | neg | d |
| Q84 | L207 | | neg | i |
| A81 | K208 | | **neg** | i |
| *A81 | T209 | | **neg** | i |
| A81 | M210 | | *pos* | i |
| A81 | T289 | | *neg* | i |
| Q84 | T289 | | **neg** | i |
| S83 | D357 | 1st | **neg** | d |
| S83 | Q358 | 2nd | neg | d |
| A81 | L369 | 2nd | *pos* | d |
| S83 | L369 | 2nd | pos | i |

Analysis of the pH dependence showed that the ΔF was more pH-sensitive than was the current activation in the majority of the palm triple mutants (*Figure 6—figure supplement 3*). There was no trend of mutants with faster ΔF kinetics having ΔF $pH_{50}$ values closer to their $pH_{50}$ of current activation (*Figure 6—figure supplement 4*).

## Complex intersubunit distance changes are associated with channel opening

The VCF analysis indicated two intrasubunit and four intersubunit distance changes occurring with fast kinetics, before or with channel opening (Figure 8B, *Video 1*). The two intrasubunit distance changes indicated an approaching between the distal β5-β6 loop (Pro205, Leu207) and the β1−β2 linker (Ala81, Gln84). The intersubunit distance changes indicated an approaching between Ala81 of

**Table 2.** Distances between Cys / quencher pairs.

The distances were calculated between the β-carbon residues of the indicated residues from structural models of the closed (**Yoder et al., 2018** PDB code 5WKU), Mit-toxin-opened (**Baconguis et al., 2014** 4NTW) and desensitized state (**Gonzales et al., 2009** 4NYK) of hASIC1a in Chimera (**Pettersen et al., 2004**). The distances were measured in all three subunits, and the average was calculated. 'Average distance' is the average of the distance measured in the structural models corresponding to the three functional states. The differences of these distances between two functional states are presented in a way that negative values indicate a shorter distance in open relative to the closed, and desensitized relative to the open conformation.

| Residue1 | Residue2 | Distance (from structure) | | |
|----------|----------|---------|------------|------------|
| | | Average | Difference | |
| | | | open- | desensitized |
| | | | closed (Å) | -open (Å) |
| A81 | Y417 | 8.4 | 0.3 | −1.5 |
| S83 | Y417 | 10.9 | 0.0 | −2.7 |
| Q84 | Y417 | 10.5 | 0.0 | −0.4 |
| A81 | P205 | 23.5 | 0.1 | −0.5 |
| S83 | P205 | 17.4 | −0.4 | −0.4 |
| Q84 | P205 | 13.0 | −0.1 | −0.6 |
| Q84 | R206 | 11.0 | −0.2 | 0.0 |
| A81 | L207 | 19.6 | 0.4 | −2.1 |
| Q84 | L207 | 10.6 | −0.2 | 0.7 |
| A81 | K208 | 15.0 | 0.3 | −1.3 |
| A81 | T209 | 13.1 | −0.2 | −3.1 |
| A81 | M210 | 11.7 | −0.4 | −3.0 |
| A81 | T289 | 8.5 | 0.2 | 2.2 |
| S83 | T289 | 12.4 | −2.0 | −0.9 |
| Q84 | T289 | 16.2 | −0.8 | −1.7 |
| S83 | D357 | 11.5 | 3.6 | −1.7 |
| S83 | Q358 | 7.1 | 4.5 | −0.5 |
| S83 | E359 | 8.8 | 3.0 | −1.0 |
| A81 | L369 | 10.9 | −2.4 | −2.1 |
| S83 | L369 | 18.2 | −3.7 | −0.9 |

the β1−β2 linker and 'Leu369 of the palm of a neighboring subunit, as well as between Ser83 and 'Thr289, which is located close to the β-turn of the β9-α4 palm-thumb loop. In addition, they indicated an approaching between Ser83 and 'Glu359, while the distance between Ser83 and 'Gln358 increased.

To provide a structural interpretation of at least some of the predicted conformational changes, an unbiased MD simulation of a structural model of hASIC1a was carried out. Molecular dynamics simulations have been used extensively to study biomolecular processes at atomic resolutions, including ion channel activation (**Bignucolo et al., 2015**; **Kasimova et al., 2019**; **Lynagh et al., 2017**; **Hadden et al., 2018**). The MD simulation was conducted for a duration of 740 ns with hASIC1a in the resting state, embedded in a membrane and protonated according to a pH of 5, based on pKa calculations (Materials and methods). These conditions mimic an acidification administered to a closed channel. They would therefore initiate conformational changes leading to channel opening. In this short MD simulation, it may be possible to observe initial transitions that would ultimately lead to channel opening. To identify possible distance changes between residues at the level of the side chains, the evolution of these distances over time was plotted for the six pairs identified

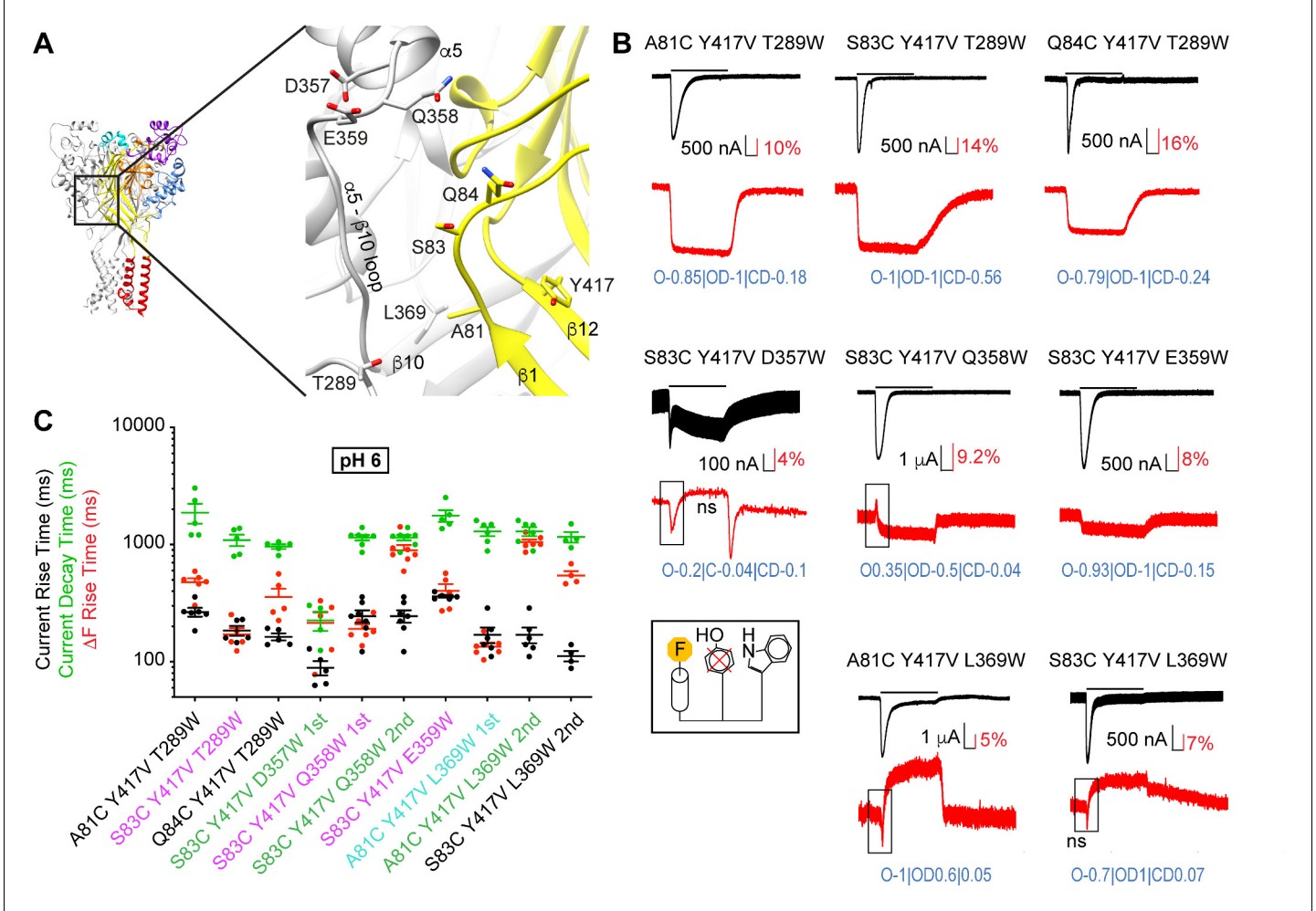

**Figure 6.** Fast structural rearrangements in the palm-thumb loops. (**A**) Structural image of ASIC1a (left) with a zoom on the palm (right), showing the position of mutations studied in this figure. (**B**) Representative current and fluorescence traces of the triple mutants. Conditioning pH7.4 was used, and mutants were stimulated by pH6 for the duration indicated by the horizontal bars. The black frames in some ΔF traces highlight the first ΔF component. 'ns' indicates that this ΔF component was potentially non-specific (**Supplementary file 3**). Proportionality factors of the corresponding simulated traces are indicated in blue. (**C**) Current RT and decay time and ΔF RT obtained at pH6, n = 4–7. The color of the labels of the mutants indicates that the ΔF onset kinetics are correlated with current appearance (purple) or decay (green) or are faster than current appearance (cyan; Materials and methods). Source data are provided in the file **Figure 6—source data 1**.

The online version of this article includes the following source data and figure supplement(s) for figure 6:

**Source data 1.** Intersubunit triple mutations palm.

**Figure supplement 1.** Fast ΔF kinetics of intersubunit palm triple mutants at pH6.5 and 5.5.

**Figure supplement 2.** Model-generated ΔF patterns for four mutants of **Figure 6**.

**Figure supplement 3.** pH dependence of palm mutants.

**Figure supplement 4.** Correlation between the ΔF - current pH$_{50}$ difference and ΔF kinetics.

by VCF in the closed-open transition (**Table 1**). Two changes were effectively observed in the trajectories (**Figure 7A–B**, **Video 2**), an increase of the distance between Ser83 and 'Gln358 from initially 5.7 Å (average during the first 20 ns of simulation, for three subunits) to 8.4 Å (average during the last 200 ns of simulation), and an approaching between Ser83 and 'Glu359 from 8.1 Å to 6.1 Å. The distances within the other four pairs did not change during this period. Snapshots at 6 and 640 ns of the simulation illustrate that the side chains of 'Gln358 and 'Glu359 exchange their orientation (**Figure 7D**). Further inspection of the MD simulations showed a continuous decrease of the distance between the side chains of 'Glu359 and Lys211, from initially 18.4 to 10.1 Å (**Figure 7C**, **Video 2**). In one of the subunit interfaces, the distance between the side chains decreased from 16.9 to 3.8 Å, a

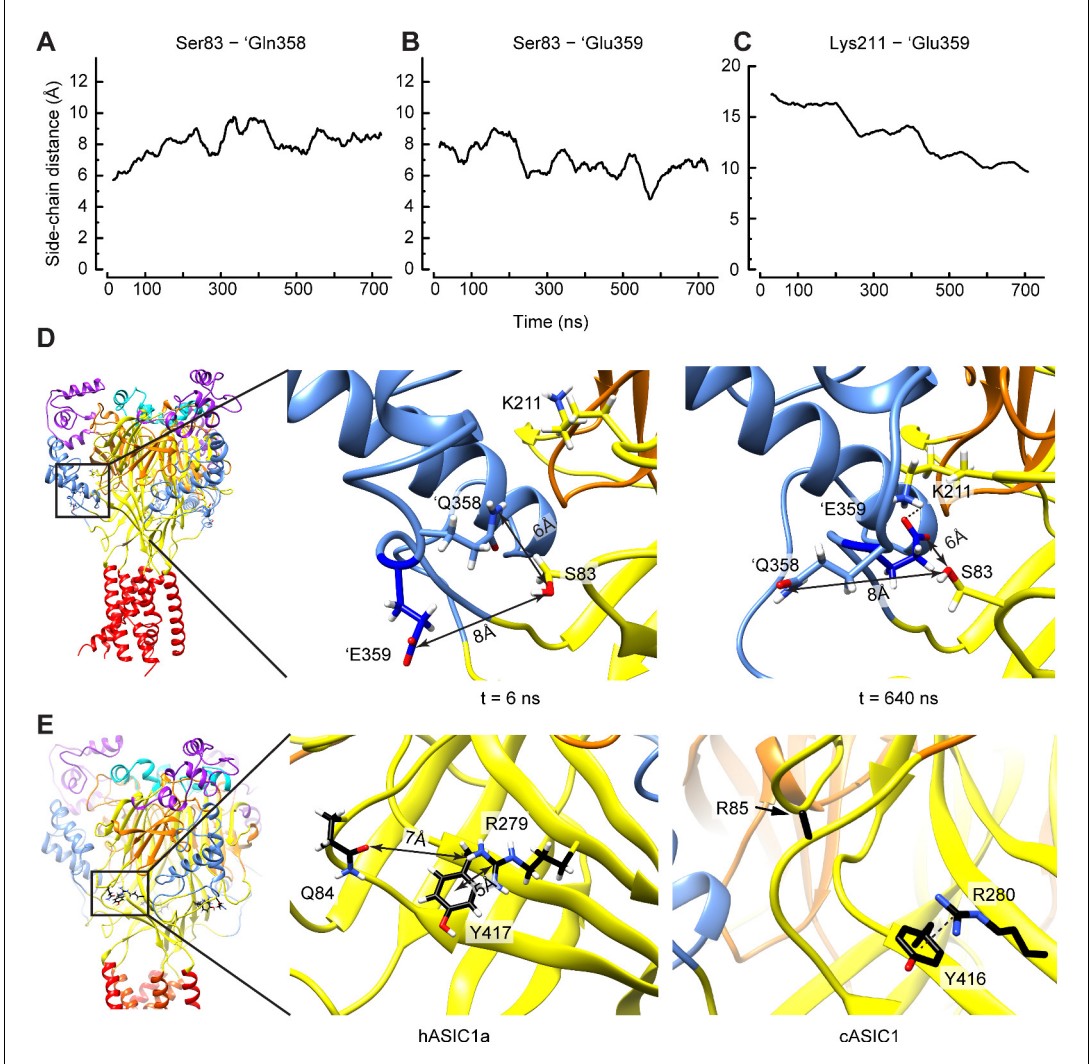

**Figure 7.** Structural interpretation of selected VCF-predicted distance changes. (**A–D**) Data and structural images based on unbiased MD simulations starting from a closed structural model of hASIC1a, in which the protonation of residues mimicked pH5. (**A–C**) Time series reporting the average (n = three subunits) of the distance between the side chains of Ser83 and 'Gln358 (**A**), Ser83 and 'Glu359 (**B**), and Lys211 and 'Glu359. (**D**) Snapshots taken at the start (t = 6 ns, left) and toward the end (t = 640 ns, right) of the simulation, highlighting the switch in side chain orientation of 'Gln358 and 'Glu359, with 'Glu359 facing the solvent at start and approaching Lys211 after a few hundreds of ns, while 'Gln358 is exposed to the solvent toward the end of the simulation. Note that the switch in side chain orientation occurred in all three subunits, while the formation of the stable salt bridge occurred in one subunit interface only. The arrows indicate the distance between the center of mass of the side chain heavy atoms (Glu: OE1 and OE2, Gln: OE1, NE2, Ser: OG). (**E**) The hASIC1a, but not the cASIC1a structure, favors the approaching between Gln84 and Tyr417 during the open-to-desensitized transition. The homologous residues in cASIC1a are Arg85 and Tyr416. Left, structural image of hASIC1a. The frame indicates the magnified area shown in the center panel. Center, molecular representation from an MD trajectory of hASIC1a showing that Tyr417 forms a cation-π interaction with Arg279. Right, in the corresponding area of the cASIC structure (PDB 4NTW), Tyr416 also forms a cation-π interaction with Arg280 (homologous to hASIC1a-Arg279). However, in this case, Arg85 is not likely to approach this positively charged area. Note that the side chain of Arg85, except Cβ, is absent from this open as well as from the desensitized structures (PDB code 4NYK). Electrostatic interactions are symbolized with black dashed lines. Source data are provided in the file *Figure 7—source data 1*.

The online version of this article includes the following source data for figure 7:

**Source data 1.** Molecular Dynamics simulations.

distance that was maintained during the last 100 ns of simulation and is compatible with the formation of a salt bridge. These observations suggest that 'Glu359 might engage transiently in electrostatic interactions with Lys211 during the first steps of the channel activation. In a recent study we observed that the distance between 'Glu359 and the neighboring palm domain was correlated with

the pH dependence of activation (*Bignucolo et al., 2020*), suggesting that facilitating the decrease of this distance would enhance the activation probability.

## An approaching between subunits associated with desensitization

The VCF results predicted 16 specific distance changes between residue pairs with ΔF kinetics that were either intermediate between those of opening and desensitization or correlated with desensitization (*Table 1*). Of these, two pairs were not considered for the structural interpretation, Ala81-Pro205 because of the high distance (~24 Å), and Ala81-Thr209 because the A81C/Y417V/T209W current did not desensitize. Some of the remaining 14 predicted distance changes are related to each other. By grouping them, three principal intra- and three main intersubunit distance changes can be distinguished (*Figure 8C*, *Video 3*). The intrasubunit distance changes include (1) an approaching between the distal β5-β6 loop and the β1-β2 linker (four pairs), (2) an approaching between the β1-β2 linker and Tyr417 of the palm β-strand 12 (3 pairs) and (3) an increase in distance between Ala81 and Met210 (proximal β5-β6 loop). The intersubunit distance changes include (1) an approaching between the β1-β2 linker and 'Thr289 (β-turn; two pairs), (2) an approaching between the β1-β2 linker and the thumb residues 'Asp357 and 'Gln358 (two pairs) and an increase in distance between the β1-β2 linker and 'Leu369 of a neighboring palm (two pairs, indicated by the second component of the VCF signals).

Three of the VCF-predicted distance changes appeared to contradict the predictions that are based on the published structures. First, the approaching between 'Thr289 and the neighboring β1-β2 linker, already observed during opening, was continued during desensitization, whereas a comparison of the crystal structures indicates an increase of the A81-'T289 distance upon desensitization. Thr289 is located close to the β-turn in the β9-β4 loop. In cASIC1a, Thr289 is replaced by an Asp residue, which could engage in different interactions with the neighboring Lys292 (Lys291 in hASIC1a). In addition, this loop is relatively poorly resolved in the crystal structure, since residues 297 and 298 (hASIC1a-296 and −297) are missing, thus possibly decreasing the reliability of some structural distances. Secondly, the distance increase between Ala81 and 'Leu369 of the neighboring palm contradicts the structure predictions. Comparison of the human and chicken ASIC1a structures does not reveal any obvious explanation for the contradiction related to the Ala81 - 'Leu369 pair. Leu369 is located within a narrow pocket of hydrophobic residues. Since Ala81 is located close to Leu415 and Asn416 that undergo an isomerization during desensitization (*Baconguis and Gouaux, 2012*; *Rook et al., 2020*), it may be involved in conformational changes linked to this isomerization. Thirdly, VCF predicted a distance increase between Ala81 and Met210 of the proximal β5-β6 loop, while the comparison of the crystal structures indicated a 3 Å decrease of this distance. One difference between human and chicken ASIC1a in the proximity of Met210 is the residue at position 208, which is a Lys in human, and Ile in chicken ASIC1a. In both isoforms, an Arg residue is located at two positions more distal on the β5-β6 loop, hASIC1a-Arg206, and cASIC1a-Arg207. Chicken ASIC1a contains an Arg residue in the β1-β2 linker at the position homologous to hASIC1a-Gln84. Thus, in cASIC1a, there is one basic side chain each in this range of the β1-β2 linker and β5-β6 loop (Arg85 and Arg207), while in hASIC1a, the two basic residues are located on the β5-β6 loop (Lys208 and Arg206). It is not clear whether these amino acid substitutions may be at the origin of the difference between VCF- and structure-based predictions regarding the movement of the proximal β5-β6 loop relative to the β1-β2 linker.

The open and desensitized structural models show no difference in distance between the β1-β2 linker and the distal residues Pro205 and Arg206 of the β5-β6 loop, nor between Gln84 and Tyr417 (*Table 2*), while VCF predicts in these cases an approaching. These differences arise likely from the fact that the residue homologous to hASIC1a-Gln84 is an Arg in cASIC1a, as mentioned above. In cASIC1a, the approaching between the β1-β2 linker and the distal β5-β6 loop would bring Arg85 and Arg207 (homologous to hASIC1a-Arg206) close to each other, which is unlikely to occur due to the repulsion between basic residues (*Figure 7E*). In both cASIC1a and hASIC1a open structures, Arg280 (hASIC1a-Arg279) forms a cation-π interaction with Tyr416 (hASIC1a-Tyr417), since the guanidium group is stacked over and located within 6 Å from the aromatic ring. In hASIC1a, Gln84, which is located within a loop and exposed to the solvent, can easily approach the Arg279 side chain (*Figure 7E*, center panel). While this distance seems too large to support stable electrostatic interactions, the structural comparison shows that this residue does not constitute a barrier like Arg85 in cASIC1 (*Figure 7E*, right panel).

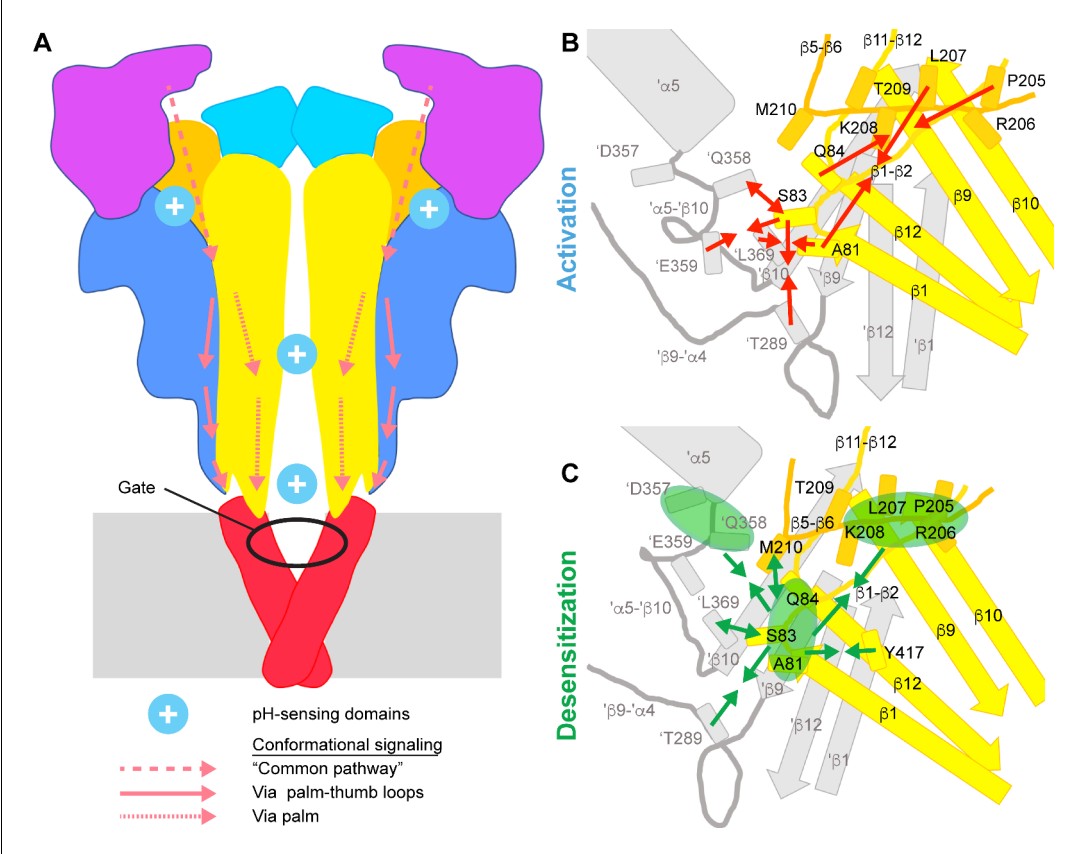

**Figure 8.** Conformational changes leading to activation and desensitization. (**A**) Activation signaling pathways in ASIC1a. Cartoon of the ASIC structure, in which the predicted pH-sensing regions are highlighted (only two of the three acidic pockets are indicated) and hypothesized signaling pathways for channel activation from distal regions to the gate are indicated; dashed arrows, common pathway, dotted arrows; palm pathway; solid arrows, palm-thumb loop pathway, with the β-turn interacting with the upper end of the TM1. (**B**) and (**C**), Cartoons indicating the conformational changes in the palm and palm-thumb loops predicted by VCF to occur with ASIC activation (**B**) and desensitization (**C**). The lower palm domain of one subunit is shown in yellow, structural elements of a neighboring subunit are shown in gray. Arrows that point toward each other indicate an approaching between residues, while arrows pointing away from each other indicate an increase in distance. In C, the arrows indicate in some cases not the distance changes between individual residue pairs, but between groups of residues. The three groups indicated by green ellipses represent residues of the β1-β2 linker (Ala81, Ser83, Gln84), residues of the distal β5-β6 loop (Pro205, Arg206, Leu207, Lys208) and residues in proximity of the thumb α5 helix ('Asp357, 'Gln358). The individual distance changes during the closed-open and open-desensitized transitions are illustrated in the **Videos 1** and **3**, respectively.

## Discussion

We show here that conformational changes related to channel opening occur at the same time in the extracellular pore entry and in more distant sub-domains, such as the finger, acidic pocket, palm and thumb-palm loops, supporting the view that pH sensing at peripheral sites contributes to ASIC activation. Our VCF analysis indicates that fast conformational changes tend to occur more in the two thumb-palm loops, while slower changes occur in the palm. The lower thumb-palm loop contains the β-turn that interacts with the TM1 and may, via this interaction, lead to the opening of the channel gate (**Figure 8A**).

### Association of ΔF signals with functional transitions

To test for an association between ΔF signals and functional transitions, the kinetics of current and ΔF signals were compared. This showed, comparable to many other VCF studies, that some ΔF signals are timely associated with defined functional transitions, while many mutants showed intermediate kinetics, likely because they report conformational changes that are not directly associated with the channel gate but may be part of processes that precede and lead to the functional transition. In addition to this analysis, we have applied kinetic modeling that assumed that the fluorescence signal

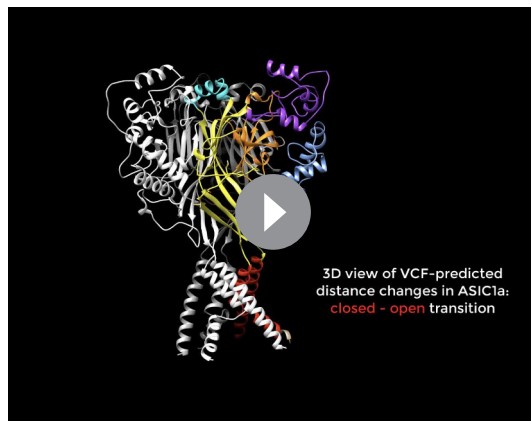

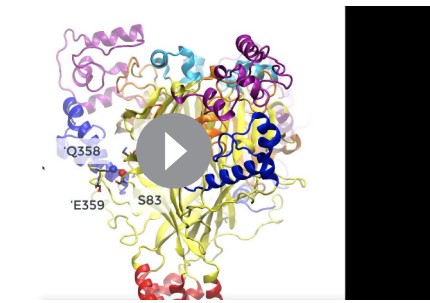

**Video 1.** Animation illustrating the VCF-predicted distance changes in the closed-open transition. Animation based on a structural model of hASIC1a in the closed state (PDB 5WKU). Red solid lines indicate an approaching between the two residues of a given pair, while dashed lines indicate a distance increase between them. Note that the animation does not show the conformational changes; it just indicates where the distance changes are predicted to occur. The timing in the animation is not indicative of the timing of the distance changes in the represented pairs.

https://elifesciences.org/articles/66488#video1

**Video 2.** Animation showing a Molecular Dynamics simulation that highlights the flip in rotameric positions of 'Gln358 and 'Glu359. The video shows the Molecular Dynamics trajectory, started from a closed state model of ASIC1a, analyzed in Figure 7A–D. The side chain and backbone atoms of residues Ser83, Lys211, 'Gln358 and 'Glu359 are shown, with the carbon, oxygen, and nitrogen atoms colored black, red and blue. Whenever a heavy atom of 'Gln358 or 'Glu359 is within a distance of $\leq 3$ Å from a heavy atom of Ser83 or Lys211 during the simulation, the heavy atoms of the interacting residues are shown as spheres. The total MD simulation time covered by this video is 740 ns. In the presented subunit interface, the orientation exchange of 'Gln358 and 'Glu359 was followed by the subsequent formation of a salt bridge between Lys211 and 'Glu359.

https://elifesciences.org/articles/66488#video2

is a linear combination of the probabilities of finding the ASICs in individual functional states. Although care must be taken when assigning predefined fluorescence levels to given states, and although this simple model does not consider any intermediate states and is based on WT ASIC1a, it was able to reproduce most of the measured ΔF signals rather well. The scaling factors in these models suggested for most mutants that the open and open-desensitized states were the principal contributors to the ΔF signal.

For the interpretation of VCF data, potential limitations have to be taken into consideration. VCF is carried out with mutants, and although mutants are selected that have similar properties as the WT, there are differences with regard to the kinetics and the ligand concentration dependence. In this study, ΔF kinetics were directly compared to the current kinetics, therefore the conclusions were not influenced by kinetic differences of the mutants to the WT. VCF measurements were done in the pH range 6.5–5.5 and were qualitatively similar over this range. Since for the vast majority of the tested mutants, the $pH_{50}$ shift relative to the WT was <0.5 pH units, these shifts do not influence the conclusions.

Experiments with ultrarapid perfusion systems have shown that ASICs activate with time constants of the order of 10 ms (*Bässler et al., 2001*; *Sutherland et al., 2001*; *Alijevic et al., 2020*). In the measuring chamber used in this study, the RT of solution change was ~300 ms (Materials and methods). The measured kinetics were therefore limited by the speed of solution change, and since the measured RTs of many mutants were faster than 300 ms, peak amplitudes were probably reached before the solution was completely exchanged (Materials and methods). In each experiment and for each condition, the kinetics of ΔF and current were directly compared, providing thus correct conclusions of the ΔF kinetics relative to the current kinetics, and therefore a valid attribution of the ΔF signals to functional transitions. In the interpretation of the data, it has to be considered that the absolute speed of the measured kinetics was limited by the speed of the solution change, and that the actual pH of the measurement of the kinetics was slightly more alkaline than that of the test solution.

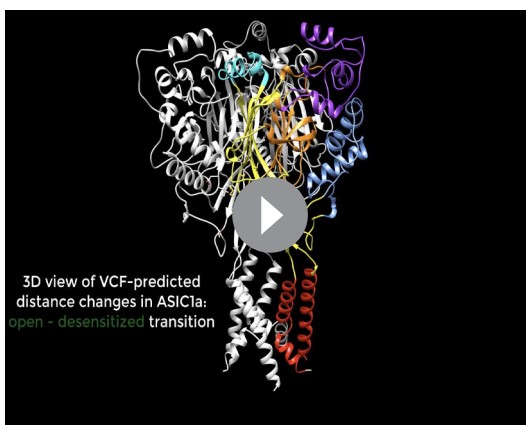

**Video 3.** Animation illustrating the VCF-predicted distance changes in the open-desensitized transition. Animation based on a structural model of hASIC1a in the open state (PDB 4NTW). Green solid lines indicate an approaching between the two residues of a given pair, while dashed lines indicate a distance increase between them. Note that the animation does not show the conformational changes; it just indicates where the distance changes are predicted to occur. The timing in the animation is not indicative of the timing of the distance changes in the represented pairs.
https://elifesciences.org/articles/66488#video3

## Dissociation of ΔF and current pH dependence

The pH dependence of the ΔF signal was in most mutants, even mutants associated with activation, shifted to more alkaline values than the pH dependence of current activation, and closer to the pH dependence of SSD. Such a shift has been observed in previous studies (*Vullo et al., 2017*; *Bonifacio et al., 2014*). An alkaline shift was observed here even with the non-desensitizing mutant A81C/Y417V/T209W, thus it does not indicate a link to desensitization. It is thought that in ASICs, as in other ion channels, conformational changes in different parts of the channel will eventually lead to the opening of the channel pore. The pH dependence of the current reflects the pH dependence of the ultimate steps in this pathway. It is possible that conformational changes that do not belong to these ultimate steps have a different pH dependence. The $pH_{50}$ values depend on the two ends of the scale, the most alkaline pH at which a ΔF signal or a current is detected, and the pH at which its increase saturates. ΔF signals appear to saturate in many mutants at less acidic pH than the current. The observed alkaline shift in the pH dependence of fluorescence is reminiscent of the hyperpolarization shift of the fluorescence relative to the current voltage dependence of $K^+$ channels (*Mannuzzu et al., 1996*; *Cha and Bezanilla, 1997*). A divergence of the concentration dependence between the ΔF signals and the current was also observed for other ligand-gated channels for fluorescence signals associated with activation (*Dahan et al., 2004*; *Pless and Lynch, 2009*).

## Structure-derived rearrangements in the palm and wrist

Structure comparisons indicate that ASIC opening is associated with a collapse of the central vestibule – enclosed by the upper parts of the lower palm β sheets – and at the same time an expansion of the lowest end of the lower palm, together with an iris-like opening of the gate located in the transmembrane domains (*Yoder et al., 2018*). These rearrangements are accompanied by a displacement of the β-turns. The collapse of the central vestibule continues during the open-desensitized transition. It was proposed that during desensitization the pore would be uncoupled from this continued movement, allowing the gate to relax back to a non-conducting position (*Yoder et al., 2018*). A limitation of the available structures is the fact that the open structures of ASICs were not opened by acidic pH, but by gating-modifying toxins (*Baconguis and Gouaux, 2012*; *Baconguis et al., 2014*; *Dawson et al., 2012*). Since the opening was not induced by protonation, the conformation of some channel parts may differ from that of $H^+$-opened ASICs, as discussed (*Yoder and Gouaux, 2020*).

## Structural interpretation of VCF with fluorophore-quencher pairing

VCF with fluorophore-quencher pairing indicates whether the residues of a given pair move toward, or away from each other during a defined transition. When interpreting VCF fluorophore-quencher pairing data, one needs to be aware of the following limitations. VCF does not indicate the distance change in absolute terms between the two partners, it does not indicate which of the two partners moves, and since fluorophores are large molecules, the actually detected distance change may occur at a position slightly besides the introduced Cys residue. In addition, ΔF signals can also be caused by a reorientation of side chains without movements of the backbone.

## Conformational changes in the proximity of the β1-β2 linker during channel opening and desensitization

The VCF experiments indicate an overall approaching between structural elements at the level of the palm during activation, as illustrated in *Figure 8B* and *Video 1*. This involves movements of the β1-β2 linker relative to the distal β5-β6 loop of the same subunit, relative to 'Leu369 of a neighboring palm, and to 'Thr289 of the neighboring β-ball. Pairing of Ser83 with 'Gln358 and 'Glu359, located both on the 'α5-'β10 loop very close to the lower end of the thumb α5 helix, indicated an approaching between Ser83 and 'Glu359 and a distance increase between Ser83 and 'Gln358. Our unbiased MD simulation of hASIC1a suggests that 'Gln358 and 'Glu359 switch their side chain orientations (*Video 2*), and that there is an approaching between 'Glu359 and Lys211, which resulted at one of the three subunit interfaces in the formation of a stable salt bridge.

These movements are followed by conformational changes predicted from the VCF kinetics to occur between opening and desensitization, or with desensitization (*Figure 8C* and *Video 3*). Within the palm of each subunit, several fluorophore-quencher pairs support an approaching between the β1-β2 linker and (1) Y417 of the palm β strand 12, and (2) residues of the distal β5-β6 loop. This is accompanied by an approaching between the β1-β2 linker and 'Asp357 as well as 'Thr289 of a neighboring subunit, and a distance increase to 'Gln358 and 'L369. The generally slow conformational changes in the palm further support the tight link between the palm and desensitization, adding to the previously demonstrated importance of the β1-β2 linker in determining the desensitization kinetics (*Coric et al., 2003*), the established role of the β11-β12 linker with whom the β1-β2 linker tightly interacts (*Springauf et al., 2011*; *Rook et al., 2020*; *Baconguis and Gouaux, 2012*), as well as the importance of the lower palm β strands (*Roy et al., 2013*; *Wu et al., 2019*) in desensitization. The approaching of 'D357 to β1-β2 linker residues may be related to slow conformational changes in the acidic pocket, which are not predicted from the structure comparison, for which there is however strong evidence from VCF experiments (*Vullo et al., 2017*).

## ASIC pH sensing and signaling toward the gate

Mutations of titratable residues in the palm, wrist and acidic pocket affect the pH dependence of ASICs, and although crystal structures predict an important role of the acidic pocket in ASIC activation (*Jasti et al., 2007*), it was shown that pH sensing in the acidic pocket is not required for the generation of transient $H^+$-induced ASIC currents (*Vullo et al., 2017*). In the lower palm, putative pH-sensing residues were identified in several β strands (*Liechti et al., 2010*; *Krauson et al., 2013*). Mutation of the wrist residue His73 to Ala shifted the pH dependence of activation to more acidic values, and simultaneous mutation of His72 and His73 to Asn suppressed pH-induced currents completely (*Paukert et al., 2008*).

We show here that activation-related conformational changes occur in the finger and in the acidic pocket. Such conformational changes can likely be transmitted down to the pore from the finger and acidic pocket via the thumb and the thumb-palm loops to the TM1 involving the interactions between aromatic residues of the TM1 and the β-turn, and/or via the central scaffold and subunit interactions to the palm and from there to the transmembrane domains (*Figure 8A*). Protonation events in the palm and wrist would affect the transmission of conformational changes initiated farther up, or may drive in part the large conformational changes of these domains, which would – due to their proximity – also change the conformation of the transmembrane domains or affect the position of the β-turns. Mutation of the interacting residues between the β-turn and the TM1 disrupts ASIC1a currents (*Li et al., 2009*), leading to a lowering of the cell surface expression, and rendering the cell surface-resident channels non-functional (*Jing et al., 2011*). Normal mode analysis had suggested a high correlation between movements of the β-turn and the upper TM1 (*Yang et al., 2009*), highlighting together with the experimental data a possible importance of the β-turn for ASIC activation.

In conclusion, we show here that upon extracellular acidification, fast conformational changes occur simultaneously in different ASIC domains, consistent with the existence of multiple protonation sites. Analysis of the kinetics and the direction of conformational changes in the palm and palm-thumb loops highlights rapid events in the palm-thumb loops, which may therefore constitute a preferred communication pathway between protonation sites and the channel gate in the context of activation.

# Materials and methods

## Key resources table

| Reagent type (species) or resource | Designation | Source or reference | Identifiers | Additional information |
|---|---|---|---|---|
| Recombinant DNA reagent (DNA plasmid) | psD5-hASIC1a | PMID:16282326 | | For in vitro transcription |
| Recombinant DNA reagent (DNA plasmid) | psD5-hASIC1a-A81C | This paper | | hASIC1a containing mutation A81C for in vitro transcription; all mutants used in this study were prepared with the same approach. |
| Commercial assay or kit | KAPA HiFi HotStart PCR polymerase | Roche Diagnostics | KK2501 | PCR |
| Commercial assay or kit | mMESSAGE mMACHINE SP6 | ThermoFisher | AM1340 | in vitro transcription |
| Sequence-based reagent | Primers for mutagenesis | Microsynth | | See *Supplementary file 4* |
| Chemical compound, drug | 3-Maleimidopropionic acid | Bachem | 4038126.0001 | Blocking of free cysteines on the oocyte |
| Chemical compound, drug | CF488 maleimide | Biotium | 92022 | Fluorophore used in VCF |
| Chemical compound, drug | AlexaFluor 488 Maleimide | ThermoFisher | A10254 | Fluorophore used in VCF |
| Software, algorithm | HEKA Chartmaster and Fitmaster | Harvard Bioscience | RRID:SCR_016233 | Electrophysiology software, https://www.heka.com |
| Software, algorithm | Graphpad Prism | Graphpad Software | RRID:SCR_002798 | Data and statistical analysis, preparation of graphs |
| Software, algorithm | Origin PRO software | OriginLab Corp, Northampton, USA | RRID:SCR_002815 | Data analysis |
| Software, algorithm | UCSF Chimera | Resource for Visualization and Informatics | RRID:SCR_004097 | Structure analysis, distances measurements |
| Software, algorithm | GROMACS | Resource for MD simulations http://www.gromacs.org | RRID:SCR_014565 2018.6 | Simulations |
| Software, algorithm | VMD | Resource for MD Visualization and calculations | RRID:SCR_001820 1.9.3 | Structure and trajectory analysis |
| Software, algorithm | R RStudio | https://www.r-project.org/ | RRID:SCR_001905 1.3. | Statistics and data analysis |
| Software, algorithm | CHARMM GUI | http://www.charmm-gui.org/ | RRID:SCR_014892 | Construction of the systems of simulations |
| Software, algorithm | SWISS-MODEL | https://swissmodel.expasy.org/ | RRID:SCR_018123 | Homology modeling |

## Molecular biology

The human ASIC1a cDNA construct (*García-Añoveros et al., 1997*) was cloned into a pSP65 vector containing 5'- and 3'- untranslated sequences for expression in *Xenopous* oocytes. As reported recently, this construct contains the mutation G212D (*Vaithia et al., 2019*). Point mutations were introduced by site-directed mutagenesis using KAPA HiFi HotStart PCR polymerase (Roche Diagnostics, Rotkreuz, Switzerland). All mutations were verified by sequencing (Synergen Biotech). Complementary RNAs were synthetized in vitro using the mMESSAGE mMACHINE SP6 kit (Thermofisher).

## Oocyte expression

Surgical removal of oocytes was carried out as described previously (*Liechti et al., 2010*). Healthy oocytes of stages V and VI were collected from adult female *Xenopous laevis* in accordance with the Swiss federal law on animal welfare and approved by the committee on animal experimentation of the Canton de Vaud. Oocytes were injected with 1–50 ng of cRNA encoding hASIC1a WT and

mutants. Oocytes used for VCF experiments were incubated after cRNA injection for 1 hr in Modified Barth's Solution (MBS) containing 10 mM 3-maleimidopropionic acid (Bachem) to modify free cysteine residues of proteins natively expressed on the cell membrane, and then maintained at 19°C in MBS, composed of (mM): 85 NaCl, 1 KCl, 2.4 NaHCO$_3$, 0.33 Ca(NO$_3$)$_2$, 0.82 MgSO$_4$, 0.41 CaCl$_2$, 10 HEPES, and 4.08 NaOH.

## Electrophysiology

Two-electrode voltage clamp (TEVC) and voltage-clamp fluorometry (VCF) experiments were conducted at room temperature (20–25°C) 1–2 days after cRNA injection. All oocytes used for TEVC and VCF experiments had been previously labeled in the dark with 5 µM CF488 maleimide (Biotium) or AlexaFluor488 maleimide (ThermoFisher) at room temperature for 15 min. Oocytes were placed in a RC-26Z recording chamber (Warner Instruments) and impaled with glass electrodes filled with 1M KCl, and continuously perfused by gravity at a rate of 5–15 mL/min. A specially designed chamber was used to measure the kinetics of the fluorescence changes (ΔF) and current from about the same oocytes surface, as described previously (*Vullo et al., 2017*). In this chamber, the solution flows under the oocyte (*Figure 1C*). Macroscopic currents were measured at a holding potential of −40 mV with a TEV-200A amplifier (Dagan Corporation). Data were recorded with Chartmaster software (HEKA Electronics) at a sampling rate of 1 kHz. Standard recording solutions contained (mM): 110 NaCl, 2 CaCl$_2$, and 10 HEPES for pH ≥6.8. For solutions with a pH <6.8, HEPES was replaced by 10 mM MES. The pH was adjusted using NaOH and HCl. For the current and ΔF measurements, channels were stimulated once every minute for 10 s (20 s for pH dependence measurements of ΔF).

The speed of solution change in the area used for the measurement of the fluorescence and the current signal of the chamber used for the kinetic measurements was determined by the same approach as described previously for a different chamber (*Bonifacio et al., 2014*). The constitutively active, Na$^+$-selective ENaC was expressed in oocytes, and oocytes were labeled before the experiment with the membrane-impermeable, pH-sensitive fluorophore 5 (6) FAM SE [5-(and-6)-carboxyfluorescein, succinimidyl ester] mixed isomers (Biotium, Chemie Brunschwig, Basel, Switzerland). In the experiments, the solution was changed from one containing K$^+$ as cation, at pH7.4 to one in which K$^+$ was replaced by Na$^+$, and whose pH was 6.0. This allowed the simultaneous determination of the solution change by fluorescence and by current measurement without the involvement of a channel activation, since ENaC is constitutively active, but not permeable to K$^+$. The rise time (RT) of the solution change was 355 ± 37 ms (current) and 296 ± 34 ms (ΔF), and the ratio RTΔF/RTI was 0.85 ± 0.07 (n = 8). These kinetics are slower than the current and ΔF measured for ASIC activation by pH in many experiments. It is known from experiments with ultrarapid perfusion systems and excised patches that the intrinsic kinetics of ASIC activation are of the order of ~10 ms (*Bässler et al., 2001*; *Sutherland et al., 2001*; *Alijevic et al., 2020*; *Vaithia et al., 2019*). Therefore, the ASIC activation kinetics are in most studies limited by the speed of the perfusion change. Our control experiments indicate that in experiments with ASICs, the current and ΔF maximum was reached before the solution change was complete, indicating that the peak was reached at a lower H$^+$ concentration than the one of the acidic solutions. Therefore, the actual acidic pH of the kinetic measurements is more alkaline than the pH of the acidic solution. Since in our experiments, relative differences between current and ΔF kinetics, measured simultaneously, are analyzed, the conclusions about the speed of the ΔF relative to the current signal, and with this the attribution to a given functional transition, is valid. It needs however to be taken into account that the absolute speed of these kinetics is limited by the kinetics of the solution change, and that the actual pH of the measurement of the kinetics is slightly more alkaline than that of the test solution.

## Fluorescence measurements

The VCF setup was equipped with an Intensilight mercury lamp (C-HGFI; Nikon) and a 40x Nikon oil-immersion objective to detect the fluorescence signal emitted by fluorophore-labeled oocytes. The optical signal was then converted into current units by a photodiode (S1336-18BQ; Hamamatsu Photonics) coupled to the head stage of an amplifier (List-EPC-7; HEKA). A low-pass eight-pole Bessel filter was used to amplify and filter the signal at 50 Hz. Changes in fluorescence intensity (ΔF) were normalized to the total fluorescence signal (F). Specificity of the fluorescence signals was assessed by exposing the oocytes to a slightly acidic pH (pH6.7) for 50 s, which puts the channels in the

desensitized state, before they were stimulated with pH6. This protocol did not generate ionic current, because the channels were desensitized before the acidification to pH6. If a Cys mutant showed a substantial fluorescence signal after application of this protocol, the signal was considered as potentially non-specific. It can, however, not be excluded that such a signal may be due to a transition between different sub-states, as for example two desensitized states. As a measure of the specificity of the signal (or a component of the signal), the ratio of the ΔF/F induced by acidification from the desensitized relative to the ΔF/F induced by acidification from the closed state was calculated (*Supplementary file 3*). The lower the ratio, the higher is the confidence that the fluorescence signal is specific.

## Kinetic model, data analysis, and statistics

To predict the time dependence of the ASIC1a probability of being in a given functional state, ASIC1a WT was modeled according to a previously published kinetic model that is based on the Hodgkin-Huxley formalism, containing an activation and a sensitization gate, and containing four functional states, closed, open, closed-desensitized, and open-desensitized (*Alijevic et al., 2020*) (https://zenodo.org/record/3909375#.YEoQDebjLy8, doi:10.5281/zenodo.3909375). Only the open state conducts current. The same model was also used as a basis for the simulation of ΔF traces (see below).

Experimental data were analyzed with the software Fitmaster (HEKA Electronics) and with Origin PRO software (OriginLab Corp, Northampton, USA). pH-response curves were fit to the Hill function: $[I = I_{max}/1 + (10^{pH_{50}}/10^{-pH})^{nH}]$, where $I_{max}$ is the maximal current, $pH_{50}$ is the pH that induces 50% of the maximal current amplitude, and nH is the Hill coefficient. Steady-state desensitization (SSD) curves were fitted with an analogous equation.

The results are presented as mean ± SEM. They represent the mean of n independent experiments on different oocytes. Statistical analysis was done with t-test where two conditions were compared, or with one-way ANOVA followed by Dunnett's or Tukey multiple comparisons test for comparison of >2 conditions for normally distributed data, and Kruskal-Wallis and Dunn's test for non-normally distributed data (Graphpad Prism 8). ΔF and current kinetics were considered as correlated for a given mutant and pH condition if the steepness of the linear regression of a plot of the rise time (as time to pass from 10 to 90% of the full amplitude) of ΔF as a function of the rise time of the current kinetics was between 0.75 and 1.33. Structural images were generated with the UCSF Chimera software (*Pettersen et al., 2004*).

## Molecular Dynamics simulations

Homology models of human ASIC1a were constructed from chicken ASIC1a which shares 90% sequence identity with its human homolog, using structures representing the closed (PDB code 5WKU) and open (4NTW) conformations (*Baconguis et al., 2014*; *Yoder and Gouaux, 2018*; *Yoder et al., 2018*) with SWISS-MODEL (*Biasini et al., 2014*). The molecular systems were constructed as done previously, inclusive the pKa calculations using the PBEQ (*Im et al., 1998*) module of CHARMM (*Bignucolo et al., 2020*). They contained ~250 or~220 1-palmitoyl-2-oleoyl-sn-glycero-3-phosphocholine (POPC) molecules in each leaflet for the closed or open state, respectively, and the total number of atoms was 281'000 and 244'000. Following the pKa calculations, the protein was protonated at pH 5.0. The GROMACS package, version 2018.6, was used to conduct simulations with the CHARMM force-field (*MacKerell et al., 1998*), versions v27 for proteins (*Mackerell et al., 2004*) and v36 for lipids (*Klauda et al., 2010*). The systems were equilibrated following the CHARMM-GUI protocol (*Jo et al., 2007*). The simulation involving the open structure was 10 ns long, because its purpose was to allow the residue side chains to adapt to their exposure to the solvent and the membrane while conserving the backbone of the crystal structure. A snapshot at t = 9 ns was taken for the center panel of *Figure 7E*. The simulation that started with the coordinates of the closed state was 740 ns long, since its aim was to capture early responses of residue side chains of a closed channel to the exposure to an acidic pH. The coordinates were taken at 2 ns interval and the measured distance values smoothed through conventional simple moving average as shown:

$D_t = \frac{1}{n}\sum_{i}^{n} d_i$ where $d_i$ are the measured individual distances, n is the strength of the filter, which was set to n = 30 in this work.

## Simulation of ΔF signals from a kinetic ASIC1a model

As one possible interpretation of the ΔF signals it was assumed that a certain degree of fluorescence was associated with each functional state (*Figure 1F–G*). To this end, the kinetic model describing the function of ASIC1a WT, described above (*Alijevic et al., 2020*), was used, and scaling factors between −1 and +1 relating the fluorescence to each of the functional states (C, closed; O, open; CD, closed-desensitized; OD, open-desensitized) were chosen to reproduce best the measured ΔF traces. Specifically, ΔF was modeled as:

$$\Delta F \sim F(C) \cdot P(C) + F(O) \cdot P(O) + F(CD) \cdot P(CD) + F(OD) \cdot P(OD),$$

where the symbol '~" denotes proportionality, P(C), P(O), P(CD) and P(OD) are the probabilities of the channel to be in the corresponding states (which evolve with time but always sum up to 1) and F(C), F(O), F(CD) and F(OD) are the corresponding constant scaling factors. The traces in *Figure 1G* show modeled ΔF traces for the case that ΔF was associated to one single functional state, or associated to either the activation or sensitization gate. The model assumes that fluorescence can be positively or negatively associated with any of the four states. We aimed to use the simplest model (least number of non-zero factors) that described the measured ΔF sufficiently well. For ease of interpretation, we included an F(C) different from 0 only if the trace could not be described with attributing values to F(C), F(O) and/or F(CD) alone. For the large majority of ΔF traces it was possible to find combinations of scaling factors that reproduced the experimental trace reasonably well (see for example *Figure 1D*). The values of F(C), F(O), F(CD), and F(OD) describing each model are noted in blue below the simulated traces, and are presented in *Supplementary file 2*. These factors were estimated empirically as follows. For some recurring, relatively simple patterns of the ΔF signal, defined rules were used to derive the factors. Simple transient signals containing in addition a sustained component (e.g. E63C) were characterized by a F(O) and F(OD). F(O) was set equal to −1 or +1 depending on the polarity of the ΔF signal, and F(OD) was chosen to reproduce the quantitatively determined $\Delta F_{sust}/\Delta F_{peak}$ ratio. Many of the measured ΔF signals were sustained, without transient component (e.g. K105C). Simulations with the kinetic model showed that the kinetics of the ΔF onset depend almost exclusively on F(O) and F(OD). The absolute value of F(OD), (abs(F(OD))) needs to be > abs(F(O)) to ensure that there is no transient component. F(OD) was set to −1 or +1 in these cases, and F(O) was based on experimentally determined kinetics at pH6.0 of the RT of the fluorescence onset (RTF) relative to the current rise time (current activation, RTAI) and the current decay time (current desensitization, RTDI) as (RTF-RTAI)/(RTDI-RTAI). This ratio was also determined with the kinetic model for a number of values of F(O), and a function relating F(O) to the (RTF-RTAI)/(RTDI-RTAI) ratio was derived from the model, and used to calculate F(O) for each mutant showing a simple sustained signal (*Supplementary file 2*, Model parameters).

For acidification from pH7.4 to pH6.0, the P(CD) is ~0 during the acidification and increases immediately upon returning to pH7.4, and decreases then slowly. This slow decrease is related to the recovery from desensitization. A slow decay of the ΔF signal upon returning to pH7.4 was observed in many mutants, as for example S83C (*Figure 4B*). In some cases, only a part of the decay is slow (I428C, *Figure 2B*). If abs(F(CD)) > abs(F(OD)), a partially sustained ΔF signal increases transiently upon returning to pH7.4 (e.g. Y71C, *Figure 2B*). F(O) and F(OD) do not influence the 'off' kinetics of the ΔF signals. To obtain values of F(CD) for each mutant, the experimental 'off' kinetics of the ΔF signal of each mutant were measured for the pH6.0-pH7.4 transition ($RTF_{off}$, *Supplementary file 1*). These kinetics were also determined for a model with F(OD)=1 and different values of F(CD), and the relationship in the model between F(CD) and $RTF_{off}$ was determined. If abs(F(OD))<1, the F(CD)/F(OD) ratio determined the 'off' kinetics (i.e. the $RTF_{off}$ in the model F(OD)=1, F(CD)=1 was equal to that of F(OD)=0.5, F(CD)=0.5). The relationship between F(CD) and $RTF_{off}$ determined in the model was then used to calculate F(CD) from the experimental $RTF_{off}$ for each mutant (*Supplementary file 2*).

## Acknowledgements

The authors thank Ivan Gautschi for expert experimental work, and Anand Vaithia and Zhong Peng for their comments on a previous version of the manuscript. This work was supported by the Swiss

National Science Foundation grant 31003A_172968 to SK. This work was supported by a grant from the Swiss National Supercomputing Centre (CSCS) under project s1037.

## Additional information

### Funding

| Funder | Grant reference number | Author |
| --- | --- | --- |
| Schweizerischer Nationalfonds zur Förderung der Wissenschaftlichen Forschung | 31003A_172968 | Stephan Kellenberger |

The funders had no role in study design, data collection and interpretation, or the decision to submit the work for publication.

### Author contributions

Sabrina Vullo, Conceptualization, Formal analysis, Validation, Investigation, Visualization, Writing - original draft, Writing - review and editing; Nicolas Ambrosio, Formal analysis, Validation, Investigation, Visualization, Writing - review and editing; Jan P Kucera, Software, Investigation, Visualization, Writing - review and editing; Olivier Bignucolo, Software, Formal analysis, Validation, Investigation, Visualization, Writing - review and editing; Stephan Kellenberger, Conceptualization, Resources, Formal analysis, Supervision, Funding acquisition, Validation, Visualization, Writing - original draft, Project administration, Writing - review and editing

### Author ORCIDs

Olivier Bignucolo http://orcid.org/0000-0003-4735-049X
Stephan Kellenberger https://orcid.org/0000-0003-1755-6198

### Ethics

Animal experimentation: Ethics Statement: The study was performed in strict accordance with the Swiss federal law on animal welfare and approved by the committee on animal experimentation of the Canton de Vaud (Permit Number: VD1462.6).

### Decision letter and Author response

Decision letter https://doi.org/10.7554/eLife.66488.sa1
Author response https://doi.org/10.7554/eLife.66488.sa2

## Additional files

### Supplementary files

• Supplementary file 1. Decay kinetics of the $\Delta F$ signal when switching back from pH6.0 to pH7.4. The decay kinetics were determined as decay time, the time to pass from 90% to 10% of the maximal amplitude ($RTF_{off}$). Note that for mutants whose signal was not sustained, the $RTF_{off}$ could not be determined.

• Supplementary file 2. Parameters of kinetic models for the simulation of $\Delta F$ traces. The modeling of the $\Delta F$ traces is described in Materials and methods. Note that F(C) was always = 0, except for S83C Y417V D357W where it was set to $-0.15$. a, F(OD) calculated from $F_{sust}/F_{peak}$ ratio; b, parameters chosen to reproduce the shape of the trace; c, F(O) calculated from the fluorescence rise time (RTF); d, F(O) and F(OD) determined from the amplitude ratio of the two components of the $\Delta F$ signal; *, experimental traces not matched by model, off kinetics are too slow in the model; & , F(C) $=-0.15$. F(CD) was in all cases except for b calculated from the RTF when the pH was switched back to pH7.4.

• Supplementary file 3. Test for intrinsic pH dependence of fluorophores. The ratio of the $\Delta F/F$ of the acidification from the desensitized state (i.e., conditioning pH6.7, stimulation pH6)/ $\Delta F/F$ of the

acidification from the closed state (pH7.4 / pH6) is shown as a percentage (n = 3–10, mean ± SEM). This ratio is not indicated here for the mutants of *Figure 1* that had previously been measured with a different perfusion system. For those mutants, the ratio was <20% (*Bonifacio et al., 2014*; *Gwiazda et al., 2015*). For most mutants, the ratio was measured using the two fluorophores Alexa Fluor 488 and CF488. The fluorophore yielding the lower ratio was subsequently used for the experiments and its ratio (as a percentage) is indicated in this table. Negative values indicate that the two protocols produced signals of opposite polarity. Mutants labeled by CF488 are marked with [C]. Alexa Fluor 488 was used for all the remaining mutants.

• Supplementary file 4. Sequences of oligonucleotides for mutagenesis. This table indicates the sequences of the oligonucleotides used for mutagenesis.

• Transparent reporting form

## Data availability

All data generated or analyzed during this study are included in the manuscript and supporting files. Source data files containing the source data for all figures are provided as supplementary files.

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
