## [Decision Letter]

**Acceptance summary:**

The molecular events that produce activation of the proton activated ASIC1a channels are beginning to be unraveled. This paper by Vullo et al., present elegant experiments that provide a framework to understand how the interaction of protons at distant sites are transmitted and converted to the opening of the pore in these channels. The authors have carried out simultaneous recordings of channel activity and conformational changes and rationalized their results in a comprehensive model of channel gating. This is a relevant work that extends the understanding of these channels, which participate in important physiological and pathophysiological processes such as pain and noxious stimuli perception.

**Decision letter after peer review:**

[Editors’ note: the authors submitted for reconsideration following the decision after peer review. What follows is the decision letter after the first round of review.]

Thank you for submitting your work entitled "Kinetic analysis of ASIC1a delineates conformational signaling from proton-sensing domains to the channel gate" for consideration by *eLife*. Your article has been reviewed by three peer reviewers including Leon D Islas as the Reviewing Editor and Reviewer #1, and the evaluation has been overseen by a Senior Editor. The following individuals involved in review of your submission have agreed to reveal their identity: John Bankston (Reviewer #2); John B Cowgill (Reviewer #3).

Our decision has been reached after consultation between the reviewers. Based on these discussions and the individual reviews below, we regret to inform you that your work will not be considered further for publication in *eLife*.

The reviewers considered this an interesting manuscript and were impressed by the large amount of work presented; however, it is the consensus that the depth of analysis is lacking and the interpretation needs to be reconsidered in the light of a more careful analysis. In particular, the sequence of the proposed conformational wave is a weak point that needs to be reinterpreted given the difficulty in assigning particular states based on fluorescence changes. We believe the required changes are substantial and require more time that allowed by *eLife*. If you decide to thoroughly address the reviewers comments, we will be willing to receive a new submission.

Reviewer #1:

The authors of this manuscript have carried out an extensive mapping of fluorescence changes associated with ASIC channel activation by pH changes. Through reaction of fluorophores at strategically introduced cysteine residues, fluorescence signals in response to rapid solution changes were recorded that had different kinetics. When comparing fluorescence and current kinetics, the authors proposed that a sequence of movements can be traced in the extracellular domains of these channels that lead to channel opening in the transmembrane domain gate.

The experiments are interesting and the interpretation that a series of conformational changes can be temporarily discerned seems to be in accord with predictions derived from available structures of ASIC in different conformations.

The paper is well written and in general, conclusions can be supported by the available experimental evidence. However, the main problem I see with this manuscript in its current form is that the different kinetic behaviors of fluorescence signals cannot be unequivocally assigned to specific channel states or gating transitions.

For example, fast fluorescence changes with the kinetics of channel opening do not necessarily mean they are associated with opening. Desensitization in these channels can occur from both open and closed states, so, fast fluorescence signals could be associated with desensitization from the closed state.

The authors suggest that the pH-independence of the ratios of fluorescence to current is indicative of florescence reflecting channel opening. This observation could also happen if fluorescence arises form desensitization coupled to channel opening. Separating closed-state desensitization from opening is not easy, but the authors should at least discuss these possibilities.

Also, please indicate if, as expected, current and fluorescence kinetics become faster with lower pH.

The observation that Tyr417 quenches the fluorescence of the Alexa dyes, leads the authors to the Trp quenching experiments. Please explain the rational for using Trp and not Tyr. One would expect that the bulkier, more hydrophobic Trp could potentially produce larger, unspecific, structural perturbations.

In subsection “The β1- β2 linker and the β5- β6 loop approach each other before desensitization”, it is argued that the fluorophores positioned at some mutants could produce signals as a result of protonation of the dyes. This argument can be applied to dyes introduced in all positions studied here. Please show control experiments indicating the ph-dependence of the fluorescence of a dye introduced in a non-dynamic position.

Related to this issue, for position in which the Alexa was paired with Trp, please show, as a control, the fluorescence of these modified positions in the absence of Trp. Presumably there is no fluorescence change in the absence of the quencher that can confound the interpretation of these experiments (Figures 3 and Figure 4).

Reviewer #2:

In this manuscript, Vullo and colleagues use VCF to look at conformational changes in many different regions of the extracellular domain of ASIC1a in order to determine which regions of the channel may be involved in pH sensing versus gate opening and to determine the directionality of a number of these conformational changes. The experiments are, overall, well executed and the data are high quality, but there are a number of issues with the analysis and interpretation that lessen my enthusiasm for the work. My specific concerns are listed below

1) Overall there is an enormous amount of data here and each figure could almost be expanded into its own more detailed paper. In the end, I think there needs to be a more thorough analysis and discussion of this data in order to understand what the take home messages might be. There is a lot of nuance in almost each labelled position, and many are just not discussed at all and I think the conclusions drawn by the authors are overly simple compared to the complexity of the data. I will just give a few examples below, but this problem persists throughout.

2) Figure 1 is used to conclude that "fast conformational changes occur simultaneously in different ASIC domains, consistent with the existence of multiple protonation sites." It is almost certainly true that lots of sites on the channel are protonated, but it’s not clear if these conformational changes are related to a conformational change that is actually associated with pH sensing and/or gating. For a few examples, T419C F rise time matches the rise time of the current rise time (Figure 1D), but the pH0.5 of the current is ~6.5 while the pH0.5 of the F is more like 6. So at pH 6 half the channels have undergone this conformational change as measured by F, but all of the channels are open. So it’s not clear to me, what this pH dependent conformational change represents. Then if you look at E63 the pH0.5 of the F is close to the pHDes0.5 and more alkaline that the pH0.5 of the current but no discussion of these data is given. The F signal might be alkaline relative to the current because ASICs are trimers and 3 subunits must undergo the conformational change at this site before the channel opens. This is analagous to the movement of 4 voltage sensing particles in the Hodgkin Huxley model which requires that the F curves in these channels be raised to the fourth power to match the current. Or maybe the conformational change is related to the entry into desensitization. But this sort of examination of the data is almost totally missing from the paper.

3) As another example, in Figure 3, A81C/Y417V/P205W and A81/Y417/M210 shows an increase in F suggesting A81 is moving away from P205 and M210. But then A81, S83 and Q84 are moving closer to 205, 205, and 209 in every other case. While this may all be true and some of the differences might be simply explained by a change or flip in rotameric position of the side chains, I think the analysis done here and the Discussion is not detailed enough to create a clear picture of what is happening at each of the regions of the channel. This is made clear in the model Figure in 5B and C in which the loop, where S83 is, is somehow simultaneously moving in opposite directions. Perhaps molecular dynamics simulations using some of the constraints from the data could help pull together a picture from this amount of data, but as it stands it’s just hard to understand the overall conclusions.

4) Not every mutant has a pH0.5 measurement. S83 alone shifts the pH0.5 to about 6 such that the channel is only half activated at the pH they use through much of the paper. Meanwhile, WT is probably 75% activated at pH 6 and A81 is 100% activated. It makes interpreting the data challenging. And if for some reason these triple mutants had larger effects on pH dependence it would even further complicate the interpretation.

Reviewer #3:

ASICs are pH sensing ion channels that have garnered interest as potential targets for treatments of a wide variety of physiological disorders and diseases. Structures are available for the channel putatively in all three observable functional states of ASICs. These highlight the structural changes that must occur during a gating cycle, but as of now, the kinetics of the various structural changes have not been established. Vullo et al., use VCF to correlate the conformational changes in various regions of the ASIC extracellular domain to functional changes during channel gating to establish an understanding of the order of events leading to channel opening then desensitization. Furthermore, they utilize fluorophore-quencher pairs to establish directionality to conformational changes. Overall, the manuscript is well written, and the experiments are expertly conducted, and these findings should help the field better understand ASIC channel gating. I have only a few concerns which I have listed below:

The plots with the split y-axes are very difficult to interpret. Perhaps the authors could instead plot the y-axis on the log scale so that it is easier to accommodate all the data on a single plot. Log time scales are probably more appropriate for viewing time constants, which would be analogous to the RT used here, anyway. As it is, Figure 1D, Figure 3C, Figure 4C, Figure 1—figure supplement 1A,B, Figure 3—figure supplement 2B, and Figure 4—figure supplement 1A,B are not legible.

In the subsection "Fast conformational changes in ASIC domains that are distant from pore," fluorescence changes observed are attributed to movements of the labeled site. Later authors show that (at least in some cases) it is movement relative to a quencher group located elsewhere in the channel that causes fluorescence change (subsection “The β1- β2 linker and the β5- β6 loop approach each other before desensitization”). And others, the movement is attributed to the quencher and not the fluorophore (subsection “Detection of fast conformational changes in palm-thumb loops”). It might be helpful to add to the limitations section a brief discussion about uncertainty regarding whether it is the fluorophore or quencher moving. Additionally, it would be helpful if the authors included the distance range where fluorophore-quencher pairs are sensitive to movement.

An additional limitation for VCF that is mentioned in subsection “Slow approaching between the palm β1-β2 linker and β12 strand” that is not mentioned in the Discussion is that a change in orientation could produce a fluorescence change without change in distance.

Studies on kinetics of ligand-gated processes are generally restricted by the rate of solution exchange of the setup. This study somewhat gets around the issue because authors are comparing rate of the fluorescence changes relative to changes in current amplitude that is measured simultaneously. It would be helpful to have the approximate rate of solution exchange provided to compare with the kinetics of the changes in current or fluorescence.

Fluorescence changes observed using pH jump from 6.7 to 6 are deemed non-specific. Is there any evidence that there are not structural changes occurring during this change other than the lack of currents elicited? Is it possible there are multiple desensitized states and this stimulus just causes change from one desensitized state to another? The attribution of nonspecific components in the change in fluorescence to protonation of the fluorophore seems less plausible given it is not widely observed but only occurs in a couple mutants. I think the use of this protocol is a very nice control but may not completely eliminate specific signals.

[Editors’ note: further revisions were suggested prior to acceptance, as described below.]

Thank you for submitting your article "Kinetic analysis of ASIC1a delineates conformational signaling from proton-sensing domains to the channel gate" for consideration by *eLife*. Your article has been reviewed by four peer reviewers including Leon D Islas as the Reviewing Editor and Reviewer #1, and the evaluation has been overseen by Kenton Swartz as the Senior Editor. The following individuals involved in review of your submission have agreed to reveal their identity: John Bankston (Reviewer #2); John B Cowgill (Reviewer #3).

Summary:

In this work, the authors propose an allosteric pathway for the activation of ASIC channels by pH. These channels are important substrates for disease and pharmacological targets. The authors use a combination of electrophysiology and fluorescence measurements to trace conformational changes in the channel protein. The main result is that pH-dependent changes occur in several extracellular proton-binding sites that converge on the final opening gating step. This work provides a large amount of valuable data for scientists interested in how ASIC channel gating might work. In addition, it provides a number of testable hypotheses about conformational changes that occur in the various domains of ASIC1a.

The authors have made several changes and improvements to the manuscript, in particular, the inclusion of a gating model helps the interpretation of experimental results in a coherent framework.

Essential Revisions:

1) The reviewers agree that the new MD calculations do not help support the conclusions. In particular, steered MD cannot be used as in this version of the manuscript to validate experimental results. Since the MD protocol is not unbiased, it seems the expected conformation change will always be warranted to occur. Please see the individual reviewers’ comments.

2) The consensus of the reviewers’ discussions is that the MD results do not help the manuscript, are largely arbitrary and should be removed unless substantially different MD protocols can be applied.

Reviewer #1 (Recommendations for the authors):

Regarding the SMD simulations.

I find it a little backwards when the author uses MD to validate an experimental result. It is usually the other way around, a simulation suggests a (theoretical) conformational change and it’s validated by experiment.

Although suggestive, the use of SMD is not well justified in the context that it is presented in the manuscript, as a validation of the conformational change. It seems to me that the criteria used in this simulation is not robust enough, why use a criterion of reaching a target distance in 40 ns ? How is the cut-off of 2.2 A justified?

SMD will by necessity produce a change in the direction in which the desired force is applied, so it seems that additional criteria should be used for its results to be taken as a validation of a conformational change suggested in experimental results.

Regarding Figures.

The schemes presented in Figure 5—figure supplement 2 and Figure 6—figure supplement 1 should be improved. A movie or animation should be used to summarize all this data. In the current figures, the conformational changes implied by the arrows are difficult to discern in a static picture. The authors should consider improving or eliminating them altogether.

Reviewer #2 (Recommendations for the authors):

While I still have concerns, the authors have added new data and tried to provide more framework for understanding this large volume of work. They have discussed the potential limitations in the discussion and tempered some of the conclusions. I don't have any additional suggestions for this manuscript.

Reviewer #3 (Recommendations for the authors):

If the protonation states were assigned assuming a pH of 5.3 as in the previous publication, I think this needs to be clarified in both the Results and the Materials and methods. If the protonation states were assigned assuming a pH of ~7, I think the simulations would need to be repeated at low pH. Indeed, any differences in the SMD for these pairs observed between the pH 5.3 and 7 condition could be informative,

Reviewer #4 (Recommendations for the authors):

The protocol used for the steered MD simulations is largely arbitrary:

how can one be sure that the length of the simulation is sufficient to observe the displacement? How about the force constant for the bias?

Is there any quantitative criterion that can be invoked to gauge the strength of this bias potential?

More fundamentally: since one is biasing the distance, what kind of conclusion can one draw based on the fact that the target has been reached? In other words: isn't it tautological that if I bias a distance then that distance change?

The authors should consider these concerns and make an effort to redesign (or at least provide some reasoning for) the computational protocol.

---

## [Author Response]

[Editors’ note: the authors resubmitted a revised version of the paper for consideration. What follows is the authors’ response to the first round of review.]

Reviewer #1:The authors of this manuscript have carried out an extensive mapping of fluorescence changes associated with ASIC channel activation by pH changes. Through reaction of fluorophores at strategically introduced cysteine residues, fluorescence signals in response to rapid solution changes were recorded that had different kinetics. When comparing fluorescence and current kinetics, the authors proposed that a sequence of movements can be traced in the extracellular domains of these channels that lead to channel opening in the transmembrane domain gate.The experiments are interesting and the interpretation that a series of conformational changes can be temporarily discerned seems to be in accord with predictions derived from available structures of ASIC in different conformations.The paper is well written and in general, conclusions can be supported by the available experimental evidence. However, the main problem I see with this manuscript in its current form is that the different kinetic behaviors of fluorescence signals cannot be unequivocally assigned to specific channel states or gating transitions.For example, fast fluorescence changes with the kinetics of channel opening do not necessarily mean they are associated with opening. Desensitization in these channels can occur from both open and closed states, so, fast fluorescence signals could be associated with desensitization from the closed state.The authors suggest that the pH-independence of the ratios of fluorescence to current is indicative of florescence reflecting channel opening. This observation could also happen if fluorescence arises form desensitization coupled to channel opening. Separating closed-state desensitization from opening is not easy, but the authors should at least discuss these possibilities.

We have in a previous study measured the time course of desensitization from the closed state, and based on these and other experiments we had generated a kinetic model of ASIC1a function that faithfully reproduced the ASIC currents under many conditions (PMID 32180707). In the revised manuscript we have determined from the kinetic model what the properties of a fluorescence signal that is associated with the desensitized state would be. The kinetic model used is based on the Hodgkin-Huxley formalism and contains therefore two desensitized states, CD (closed-desensitized) and OD (open-desensitized). The model predicts that a fluorescence signal associated with the CD state would have a bell-shaped pH dependence with a maximum at *~*pH6.8. The experimental determination of the pH dependence of fluorescence signals showed however an increase with acidic pH and then a saturation. If the two desensitized states are lumped together, the pH dependence shows a saturation (thus no bell-shaped curve), however the kinetics are those of the transition from the open to the desensitized state, and clearly slower than the kinetics of channel opening, indicating that rapid fluorescence changes are not associated with the desensitized state.

This is presented in Figure 3 and discussed in the text.

Regarding our statement on the pH-independence of the ratios of fluorescence to current, we agree with Dr. Islas that this is not at all a proof for a close association with opening, and we have removed this statement.

Also, please indicate if, as expected, current and fluorescence kinetics become faster with lower pH.

We observe this acceleration. In the manuscript we present the data at different pH values, however not in all cases side by side. A side-by-side comparison of fluorescence kinetics at three pH values of wrist mutants is for example shown in Figure 3D, confirming such a pH dependence. For the current, it is expected that the current appearance, but not the current decay is accelerated with acidic pH. This is for example shown in Figure 4D for three mutants of the palm.

The observation that Tyr417 quenches the fluorescence of the Alexa dyes, leads the authors to the Trp quenching experiments. Please explain the rational for using Trp and not Tyr. One would expect that the bulkier, more hydrophobic Trp could potentially produce larger, unspecific, structural perturbations.

We agree with Dr. Islas that Tyr would be a more conservative choice with less risk of structural interference. Trp is a stronger quencher, and it is for this reason that we have chosen to use Trp. A possible drawback is that the function of such mutants is different from the WT. All triple mutants except two mutants described in Figure 5

(A81C/Y417C/T209W, A81C/Y417C/M210W) showed an ASIC-typical transient current. The pH dependence of desensitization was generally not affected, while there was some variability in the pH dependence of activation. All mutants except for S83C/Y417V/D357W expressed big currents. Together, this indicates that the function of the mutants was quite close to that of the wild type. Since we compare current and fluorescence signal properties directly at the same pH, and carry out this analysis at three pH values, the shifts in pH dependence should not affect the conclusions. We discuss these aspects in subsection "Association of *D*F signals with functional transitions".

In subsection “The β1- β2 linker and the β5- β6 loop approach each other before desensitization”, it is argued that the fluorophores positioned at some mutants could produce signals as a result of protonation of the dyes. This argument can be applied to dyes introduced in all positions studied here. Please show control experiments indicating the ph-dependence of the fluorescence of a dye introduced in a non-dynamic position.

The fluorophores used in this study are indicated as being "intrinsically pH-independent". We observe with both fluorophores in many mutants that we have tested no fluorescence change at all. Such mutants are then not further used, because no information can be extracted. In some of them, the engineered Cys may not be accessible, in others however the pH change does probably not change enough the environment, which would explain the lack of fluorescence change. In the present study we observed big fluorescence signals with the single mutants A81C, S83C and Q84C (Figure 4 of the new manuscript). If on the background of each of these mutants Tyr417 was mutated to Val, the fluorescence signal disappeared almost completely. The Y417V mutation does not affect the reaction of the fluorophore derivative with the engineered Cys residues at positions 81, 83 and 84, but changed dramatically the fluorescence signal. This indicates therefore that the fluorophores have been attached, but that only a very small fluorescence change is recorded. If the fluorescence change was due to a pH effect on the fluorophore, it would not be affected by the mutation of Tyr417. We have carried out control protocols to estimate the specificity of the fluorescence signals for each mutant (Supplementary file 3) that are also discussed in the response to one of the points raised by Dr. Cowgill.

Related to this issue, for position in which the Alexa was paired with Trp, please show, as a control, the fluorescence of these modified positions in the absence of Trp. Presumably there is no fluorescence change in the absence of the quencher that can confound the interpretation of these experiments (Figure 3 and Figure 4).

These controls are those discussed above, which are described in the revised manuscript in Figure 4.

Reviewer #2:1) Overall there is an enormous amount of data here and each figure could almost be expanded into its own more detailed paper. In the end, I think there needs to be a more thorough analysis and discussion of this data in order to understand what the take home messages might be. There is a lot of nuance in almost each labelled position, and many are just not discussed at all and I think the conclusions drawn by the authors are overly simple compared to the complexity of the data. I will just give a few examples below, but this problem persists throughout.

For this new version of the manuscript, the analysis has been refined and completed. We will discuss this in the specific points raised below. We would like to stress here also that the fact that many residues are analyzed is also a strength of this study. Our conclusions are not based on only a limited number of mutants, but they rely on a large data sample. The extension of the analysis in the new manuscript leads to clearer conclusions.

2) Figure 1 is used to conclude that "fast conformational changes occur simultaneously in different ASIC domains, consistent with the existence of multiple protonation sites." It is almost certainly true that lots of sites on the channel are protonated, but it’s not clear if these conformational changes are related to a conformational change that is actually associated with pH sensing and/or gating. For a few examples, T419C F rise time matches the rise time of the current rise time (Figure 1D), but the pH0.5 of the current is ~6.5 while the pH0.5 of the F is more like 6. So at pH 6 half the channels have undergone this conformational change as measured by F, but all of the channels are open. So it’s not clear to me, what this pH dependent conformational change represents. Then if you look at E63 the pH0.5 of the F is close to the pHDes0.5 and more alkaline that the pH0.5 of the current but no discussion of these data is given. The F signal might be alkaline relative to the current because ASICs are trimers and 3 subunits must undergo the conformational change at this site before the channel opens. This is analagous to the movement of 4 voltage sensing particles in the Hodgkin Huxley model which requires that the F curves in these channels be raised to the fourth power to match the current. Or maybe the conformational change is related to the entry into desensitization. But this sort of examination of the data is almost totally missing from the paper.

As indicated, there is in most mutants a shift between the pH dependence of the current and of that of the fluorescence signal, that has already been shown in other studies. The origin of this shift is not understood. However, it is highly likely that in ASICs, as in other ion channels, conformational changes in different parts of the channel will eventually lead to the opening of the channel pore. The pH dependence of the current reflects the pH dependence of the ultimate steps in this pathway. It is well possible that conformational changes that do not belong to these ultimate steps have a different pH dependence. For both mutants cited above, the kinetics of the fluorescence signal at a given pH change are however very close to the kinetics of current appearance. One of the mutants tested in our study, A81C/Y417C/T209W (Figure 5) does not desensitize at all, however its pH50 of fluorescence is shifted from the pH50 of current activation similarly as other mutants. This further shows that the alkaline shift of the pH dependence of fluorescence does not indicate an association with desensitization. The pH50 value depends on the two end points of the curve, thus the least acidic pH where a signal (fluorescence or current) is measured on one end, and the more acidic pH where saturation of the signal increase occurs on the other end. Clearly, saturation occurs at less acidic pH for fluorescence signals than currents. The observed alkaline shift in the pH dependence of fluorescence is reminiscent of the hyperpolarization shift of the fluorescence relative to the voltage dependence of K channels (Manuzzu, 1996). A divergence of the concentration dependence between the deltaF signals and the current was also observed for other ligand-gated channels for fluorescence signals associated with activation, as for example in Dahan, 2004 and Pless and Lynch, 2009. We agree with Dr. Bankston that this discrepancy in the pH dependence needs to be better discussed. In the revised manuscript, we provide the pH dependence of almost all mutants, we plot the difference in pH50 (pH50(*D*F)-pH50(current activation)) as a function of the *D*F kinetics in Figure 6—figure supplement 4, and we discuss these shifts in subsection "Dissociation of *D*F and current pH dependence".

3) As another example, in Figure 3, A81C/Y417V/P205W and A81/Y417/M210 shows an increase in F suggesting A81 is moving away from P205 and M210. But then A81, S83 and Q84 are moving closer to 205, 205, and 209 in every other case. While this may all be true and some of the differences might be simply explained by a change or flip in rotameric position of the side chains, I think the analysis done here and the discussion is not detailed enough to create a clear picture of what is happening at each of the regions of the channel. This is made clear in the model Figure in 5B and C in which the loop, where S83 is, is somehow simultaneously moving in opposite directions. Perhaps molecular dynamics simulations using some of the constraints from the data could help pull together a picture from this amount of data, but as it stands it’s just hard to understand the overall conclusions.

As indicated by Dr. Bankston, some VCF-predicted distance changes contradict each other, and it is difficult to determine which of them are more reliable than others. We have now carried out two MD simulations, one for *D*F signals correlated with the closedopen, and one for signals correlated with the open-desensitized transitions. In these simulations we applied constraints that were based on the VCF data: For residue pairs predicted to approach each other, we applied harmonic forces to reduce their distance by 5 Å, and for those predicted to move away from each other, we applied forces to increase their distance. In such steered MD simulations, the applied force will decrease once the target distance is approached. At the end of the simulation, we thus measured the remaining difference to the target distance and the force that was still applied. By posing a threshold, we could thus select between distance changes that are likely to occur, and such that seem less probable. We used the MD simulations also to interpret seemingly contradicting predictions of the VCF experiments. Based on this additional analysis, the presentation and interpretation of the structural conclusions of the VCF data has been changed, as shown in Figure 7, Figure 8 and Figure 9 of the revised manuscript.

4) Not every mutant has a pH0.5 measurement. S83 alone shifts the pH0.5 to about 6 such that the channel is only half activated at the pH they use through much of the paper. Meanwhile, WT is probably 75% activated at pH 6 and A81 is 100% activated. It makes interpreting the data challenging. And if for some reason these triple mutants had larger effects on pH dependence it would even further complicate the interpretation.

We provide for large majority of mutants kinetic analyses at three pH conditions, pH6.5, 6.0 and 5.5. In the revised manuscript, we provide also, the pH dependence for most of the triple mutants. The fact that we did not do all the measurements is due to circumstantial reasons, since one student is no longer working in the laboratory, and the person who took over had to undergo a major surgery. The additional dataset (Figure 6—figure supplement 3) shows that, for the large majority of triple mutants the pH50 of current activation was between 5.5 and 6.3. The kinetics of the *D*F signal relative to the current signal do not much change between pH6.0 and 5.5. Therefore, these shifts in pH dependence should not affect the conclusions. The triple mutants that showed a large acidic shift in the pH dependence of activation are A81C/Y417V/T209W and S83C/Y417V/D357W. The first of these mutants has a sustained current and slow kinetics and was for these reasons not considered for any structural interpretation. In the mutant S83C/Y417V/D357W, the kinetics of the fluorescence signal were at pH6.0 and 5.5 very close to those of desensitization (at pH6.5, no current was measured).

Reviewer #3:ASICs are pH sensing ion channels that have garnered interest as potential targets for treatments of a wide variety of physiological disorders and diseases. Structures are available for the channel putatively in all three observable functional states of ASICs. These highlight the structural changes that must occur during a gating cycle, but as of now, the kinetics of the various structural changes have not been established. Vullo et al., use VCF to correlate the conformational changes in various regions of the ASIC extracellular domain to functional changes during channel gating to establish an understanding of the order of events leading to channel opening then desensitization. Furthermore, they utilize fluorophore-quencher pairs to establish directionality to conformational changes. Overall, the manuscript is well written, and the experiments are expertly conducted, and these findings should help the field better understand ASIC channel gating. I have only a few concerns which I have listed below:The plots with the split y-axes are very difficult to interpret. Perhaps the authors could instead plot the y-axis on the log scale so that it is easier to accommodate all the data on a single plot. Log time scales are probably more appropriate for viewing time constants, which would be analogous to the RT used here, anyway. As it is, Figure 1D, Figure 3C, Figure 4C, Figure 1—figure supplement 1A,B, Figure 3—figure supplement 2B, and Figure 4—figure supplement 1A ,B are not legible.In the subsection "Fast conformational changes in ASIC domains that are distant from pore," fluorescence changes observed are attributed to movements of the labeled site. Later authors show that (at least in some cases) it is movement relative to a quencher group located elsewhere in the channel that causes fluorescence change (subsection “The β1- β2 linker and the β5- β6 loop approach each other before desensitization”). And others, the movement is attributed to the quencher and not the fluorophore (subsection “Detection of fast conformational changes in palm-thumb loops”). It might be helpful to add to the limitations section a brief discussion about uncertainty regarding whether it is the fluorophore or quencher moving. Additionally, it would be helpful if the authors included the distance range where fluorophore-quencher pairs are sensitive to movement.

The fact that VCF does not tell whether the quencher or the fluorophore moves is discussed in the revised manuscript in subsection "Structural interpretation of VCF with fluorophore-quencher pairing". The distance range in which quenching between a fluorophore and a quencher such as Trp or Tyr occurs was in several studies estimated as ≤ 15 Å. This information is provided in the revised manuscript at the beginning of subsection "Conformational changes in the *β*1- *β*2 linker and the *β*5-*β*6 linker precede desensitization". As indicated in the response to the other reviewers, we have carried out Molecular Dynamics simulations to help interpreting the VCF results and overcome some of its limitations.

An additional limitation for VCF that is mentioned in subsection “Slow approaching between the palm β1-β2 linker and β12 strand” that is not mentioned in the Discussion is that a change in orientation could produce a fluorescence change without change in distance.

This aspect is also discussed in subsection "Structural interpretation of VCF with fluorophore-quencher pairing" of the revised manuscript. For some fluorophore-quencher pairs we have analyzed with the MD simulation whether the backbone distance between the residues, or only the side chain distance changes, providing thus an information that we could not deduce from the VCF analysis.

Studies on kinetics of ligand-gated processes are generally restricted by the rate of solution exchange of the setup. This study somewhat gets around the issue because authors are comparing rate of the fluorescence changes relative to changes in current amplitude that is measured simultaneously. It would be helpful to have the approximate rate of solution exchange provided to compare with the kinetics of the changes in current or fluorescence.

Dr. Islas also asked us to provide such information. We provide in the revised manuscript this information in the Materials and methods, and discuss its implication in subsection "Structural interpretation of VCF with fluorophore-quencher pairing".

Fluorescence changes observed using pH jump from 6.7 to 6 are deemed non-specific. Is there any evidence that there are not structural changes occurring during this change other than the lack of currents elicited? Is it possible there are multiple desensitized states and this stimulus just causes change from one desensitized state to another? The attribution of nonspecific components in the change in fluorescence to protonation of the fluorophore seems less plausible given it is not widely observed but only occurs in a couple mutants. I think the use of this protocol is a very nice control, but may not completely eliminate specific signals.

We were not clear enough in the description of how we interpret this test protocol. If for a mutant no fluorescence change is measured when the pH is changed from 6.7 to 6, this test tells us that the measured signal under normal conditions (from an alkaline pH to pH6) is specific. If a signal is also observed with the pH change from 6.7 to 6, it indicates that the signal may be non-specific, but as you mention, it is possible that this signal reflects conformational changes occurring for example between different desensitized states. With the reorganization of the manuscript we have moved this part to the Materials and methods, and have reformulated the passage as " Specificity of the fluorescence signals was assessed by exposing the oocytes to a slightly acidic pH (pH6.7) for 50s, which puts the channels in the desensitized state, before they were stimulated with pH6. This protocol did not generate ionic current, because the channels were desensitized before the acidification to pH6. If a Cys mutant showed a substantial fluorescence signal after application of this protocol, the signal was considered as potentially non-specific. It can however not be excluded that such a signal may be due to a transition between different sub-states, as for example two desensitized states."

[Editors’ note: what follows is the authors’ response to the second round of review.]

Essential Revisions:1) The reviewers agree that the new MD calculations do not help support the conclusions. In particular, steered MD cannot be used as in this version of the manuscript to validate experimental results. Since the MD protocol is not unbiased, it seems the expected conformation change will always be warranted to occur. Please see the individual reviewers’ comments.

As requested, we have removed the steered MD simulations in the revised manuscript.

2) The consensus of the reviewers’ discussions is that the MD results do not help the manuscript, are largely arbitrary and should be removed unless substantially different MD protocols can be applied.

As indicated above, we have removed from the manuscript all results of steered MD simulations, and the conclusions that were based on them. We have added unbiased MD simulations (Figure 7).

Reviewer #1 (Recommendations for the authors):Regarding the SMD simulations.I find it a little backwards when the author uses MD to validate an experimental result. It is usually the other way around, a simulation suggests a (theoretical) conformational change and it’s validated by experiment.Although suggestive, the use of SMD is not well justified in the context that it is presented in the manuscript, as a validation of the conformational change. It seems to me that the criteria used in this simulation is not robust enough, why use a criterion of reaching a target distance in 40 ns ? How is the cut-off of 2.2 A justified?

As indicated above (Essential revisions), we have removed the steered MD results from the revised manuscript. To answer your specific questions, (1) regarding the cutoff of 2.2A: this value was chosen arbitrarily. However, about two thirds of the studied residue pairs fitted well in either one of two categories, those in which the distance and the force decreased, and those in which both parameters stayed elevated throughout the simulation. The cutoff of 2.2 A together with a cutoff for the remaining applied force separated these two populations well. (2), regarding the duration of the simulation: the simulations had a duration of 140 ns (closed-open transition) or 170ns (open-desensitized transition), because in all cases, convergence was obtained within this time. We recorded of all residue pairs the time course of the distance change. This showed that the main part of the distance change required only a few ns. The indication of 40 ns corresponded to the last 40 ns of a simulation, after reaching convergence.

SMD will by necessity produce a change in the direction in which the desired force is applied, so it seems that additional criteria should be used for its results to be taken as a validation of a conformational change suggested in experimental results.

Although a force was applied to all involved residue pairs in these simulations, the expected changes in inter-residue distance occurred only in about half of them. This indicates that in such cases, the VCF-predicted conformational change is for structural reasons not possible. Of the conformational changes that happened in the steered MD simulations, some of them may not go in the direction in which the protonation would normally bring them. These would then be falsely validated changes. Since this part was removed from the manuscript, we have not added additional criteria.

Regarding Figures.The schemes presented in Figure 5—figure supplement 2 and Figure 6—figure supplement 1 should be improved. A movie or animation should be used to summarize all this data. In the current figures, the conformational changes implied by the arrows are difficult to discern in a static picture. The authors should consider improving or eliminating them altogether.

We agree that this aspect needed to be improved. In the revised manuscript, the former Figure 9, which has become Figure 8, shows in panels B and C an overview of the VCF-predicted conformational changes in the form of structural images with arrows indicating the conformational changes. To replace the supplementary Figure 7 and 8, we have generated animations that indicate in a pseudo-3D view the different residue pairs for which the VCF experiments with fluorophore-quencher pairing predicted distance changes. We provide two animations, one for the closed-open, and one for the open-desensitized transition. These animations do not show the structural changes (which would not be possible in this detail based on the VCF data), but they identify the residue pairs and indicate whether the distance between them increases or decreases, based on the VCF predictions.

Reviewer #2 (Recommendations for the authors):While I still have concerns, the authors have added new data and tried to provide more framework for understanding this large volume of work. They have discussed the potential limitations in the discussion and tempered some of the conclusions. I don't have any additional suggestions for this manuscript.

We thank Dr. Bankston for this overall positive evaluation of our manuscript. We are aware of the limitations of our study. We discuss them openly in the Discussion.

Reviewer #3 (Recommendations for the authors):If the protonation states were assigned assuming a pH of 5.3 as in the previous publication, I think this needs to be clarified in both the Results and the Materials and methods. If the protonation states were assigned assuming a pH of ~7, I think the simulations would need to be repeated at low pH. Indeed, any differences in the SMD for these pairs observed between the pH 5.3 and 7 condition could be informative,

Since we have removed the SMD data and conclusions based on it according to the "Essential revisions", it will not be necessary to conduct similar simulations at pH 7.

Reviewer #4 (Recommendations for the authors):The protocol used for the steered MD simulations is largely arbitrary:how can one be sure that the length of the simulation is sufficient to observe the displacement?

The analysis of the SMD trajectories provided for each residue information on the time course of the changes in distance. This showed generally a rapid change in distance, if one occurred at all. This is illustrated in Figure 7A of the previous version of the manuscript for one residue pair. The simulations lasted 140ns for those started from the closed state, and 170ns for those started from the open state. During this time, we observed a saturation of the change in distance between the two residues of pairs, and convergence of the applied force as well. This is illustrated in Figure 7B of the previous manuscript for one pair in which the target was not reached. Based on this convergence we considered that the length of the simulation was sufficient.

How about the force constant for the bias?

As indicated in the previous version of the manuscript, the force changed over time, depending on whether a movement occurred or not. At the start of the simulation, the force was 300 kJ·mol^-1^·Å^-1^. This value decreased rapidly to ~ 150kJ·mol^-1^·Å^-1^.

Is there any quantitative criterion that can be invoked to gauge the strength of this bias potential?

The strength of the bias potential was chosen arbitrarily after a few initial trials, but there is no quantitative basis for this choice. With the chosen bias potential, distance changes occurred and came to a saturation during the simulation. Together, this was taken as an indication that the chosen values were adequate.

More fundamentally: since one is biasing the distance, what kind of conclusion can one draw based on the fact that the target has been reached? In other words: isn't it tautological that if I bias a distance then that distance change?

Since forces are applied, the system will move towards the target values, but this can happen only if it is structurally possible. The fact that in about half of the tested pairs the target was reached, but not in the others, indicates that such structural barriers exist, and that the strategy succeeded in ranking the distance changes as a function of their structural likelihood. We concluded that the predicted movements in pairs in which the target was not reached are less likely to occur. Within the pairs for which the target was reached, some of the movements may correspond to those induced by protonation, while some may not, but are structurally possible. The SMD would not allow us to distinguish between the two possibilities; however, it identifies the transitions that are less likely to occur.

The authors should consider these concerns and make an effort to redesign (or at least provide some reasoning for) the computational protocol.

Since these parts have been removed from the manuscript as requested, no other changes were introduced.